# On Learning Latent Models with Multi-Instance Weak Supervision

**Kaifu Wang**
University of Pennsylvania
kaifu@sas.upenn.edu

**Efthymia Tsamoura**
Samsung AI
efi.tsamoura@samsung.com

**Dan Roth**
University of Pennsylvania
danroth@seas.upenn.edu

## Abstract

We consider a weakly supervised learning scenario where the supervision signal is generated by a transition function $\sigma$ of labels associated with multiple input instances. We formulate this problem as *multi-instance Partial Label Learning (multi-instance PLL)*. Our problem is an extension to the standard PLL problem and is met in different fields, including latent structural learning and neuro-symbolic integration. Despite the existence of many learning techniques, limited theoretical analysis has been dedicated to this problem. In this paper, we provide the first theoretical study of multi-instance PLL with possibly an unknown transition $\sigma$. Our main contributions are as follows. First, we propose a necessary and sufficient condition for the learnability of the problem. This condition nontrivially generalizes and relaxes the existing *small ambiguity degree* in PLL literature since we allow the transition to be deterministic. Second, we derive Rademacher-style error bounds based on a top-$k$ surrogate loss that is widely used in the neuro-symbolic literature. Furthermore, we conclude with empirical experiments for learning under unknown transitions. The empirical results align with our theoretical findings, exposing also the issue of scalability in the weak supervision literature.

## 1   Introduction

Consider the scenario in Figure 1, where a learner aims to learn one or more classifiers $f_1, \ldots, f_n$, each of which maps an instance $x$ to its corresponding label $y$. The learner is given training examples, each of which consists of a *vector* of $M > 1$ instances $\boldsymbol{x} = (x_1, \ldots, x_M)$, and each $x_i$ is processed by one of the $f_j$'s. Differently from supervised learning, the gold labels $\boldsymbol{y} = (y_1, \ldots, y_M)$ of $\boldsymbol{x}$ are *hidden*; instead, the learner is provided with a *weak* label $s$ which is produced by applying a *transition function*[1] $\sigma$ to the gold labels $\boldsymbol{y}$. The transition $\sigma$ itself may be unknown to the learner.

The above weak supervision setting has been a topic of active research in NLP [42, 35, 34, 32, 45, 52, 22]. Recently, it has received renewed attention as it is met in *neuro-symbolic* frameworks [14], i.e., frameworks that integrate inference and learning over neural models with inference and learning over symbolic models, such as logical theories [30, 53, 13, 59, 43, 31, 24, 27]. The benefits of this setting over architectures that approximate the $f_i$'s and $\sigma$ via an end-to-end neural model [48] are (i) the ability to reuse the latent models, something particularly useful in NLP [34, 32], (ii) higher accuracy in multiple tasks, e.g., NLP [55] and visual question answering [24], (iii) greater interpretability and (iv) the ability to encode prior knowledge via $\sigma$. Below, we present an example of our learning setting in neuro-symbolic learning.

**Example 1** (SUM2)**.** *We aim to learn an MNIST classifzer $f$. Differently from supervised learning, we use training examples of the form $(x_1, x_2, s)$, where $x_1$ and $x_2$ are MNIST digits and $s$ is*

---

[1]We use the term transition function (or transition in short) instead of simply function, due to the relationship between our setting and learning with partial labels literature [8, 9].

37th Conference on Neural Information Processing Systems (NeurIPS 2023).

*their sum. The above implies a transition function $\sigma : \{0, \ldots, 9\} \times \{0, \ldots, 9\} \to \{0, \ldots, 18\}$, s.t.*
*$\sigma(y, y') = y + y'$. The target sum $s$ restricts the space of labels that can be assigned to the input*
*pair of digits $(x_1, x_2)$, e.g., if $s = 2$, then the only combinations of labels that abide by the constraint*
*that the sum is 2 are (2,0), (1,1) and (0,2). This problem is referred to as SUM2 in literature [24, 30].*

The *transition function* $\sigma$ is not necessarily one-to-one. For instance, multiple combinations of digits can lead to the same target sum. The above leaves us with two questions to answer: *can we provide any learning guarantees for the classifiers $f_1, \ldots, f_n$?* and *what if $\sigma$ is unknown?* Despite that multiple techniques for neuro-symbolic and latent structural learning have been recently proposed, those theoretical questions remain open. Practically, for the single input case ($M = 1$), our setting can reduce to that of *partial label learning* (PLL), where each input instance is accompanied by a set of labels, including the gold one [25, 11, 5, 29, 39, 54, 57, 60]. However, a key PLL learnability assumption – that the probability a

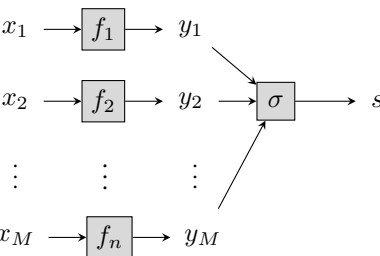

Figure 1: Multi-Instance PLL. We aim to learn the $f_i$'s given the $x_i$'s and $s$. $M$ may be different from $n$ and $\sigma$ may be unknown.

wrong label co-occurs with the gold one is always less than one [28]– is violated in our setting, rendering existing learnability results inapplicable. This is because $\sigma$ is a deterministic but *not* necessarily bijective function, as opposed to $\sigma$ in PLL which is randomized.

Due to its relevance to PLL, we refer to our learning setting as *multi-instance PLL*. Differently from prior art, e.g., [42, 35], we do not restrict $\sigma$ to specific types of functions, e.g., linear functions. Hence, we can express constraints in several formal languages, including systems of Boolean equations and Datalog [1], a language widely used in neuro-symbolic techniques [24]. Notice that many neuro-symbolic learning techniques are oblivious to the implementation of the symbolic component: abstracting the symbolic component as a transition $\sigma$ has been actually proposed as the means to compositionally integrate neural with symbolic models [43]. Furthermore, although we primarily consider deterministic transitions motivated by the neuro-symbolic and NLP literature, our results can be extended to support randomized transitions when $\sigma$ is known.

We aim to make *minimal* assumptions on the data distributions by showing learnability even under the "toughest" distributions. In addition, we provide learning guarantees under the semantic loss [56], a widely-used surrogate loss for training classifiers subject to logical theories [30, 43, 24]. To our knowledge, we are the first to provide this theoretical analysis, closing a gap in the neuro-symbolic and latent structural learning literature.

**Contributions.** Our contributions can be summarized into the following:

- We propose necessary and sufficient learnability conditions of the multi-instance PLL problem assuming all the instances are classified by a single classifier $f$ and the transition is known. We further provide a Rademacher-style error bound using a top-$k$ approximation to the *semantic loss* [56], see Section 3. Our analysis confirms the intuition that a larger $k$ decreases the empirical risk, but increases the number of samples required for learning.

- We extend the above results for the more general case where one learns multiple classifiers $\boldsymbol{f} = (f_1, \ldots, f_n)$ that classify instances from different domains, see Section 4.

- We prove learnability with a single classifier and an unknown transition function, see Section 5, and assess the validity of our theoretical results using SOTA neuro-symbolic frameworks, see Section 6.

Proofs, additional backgrounds and details on our empirical analysis are in the appendix.

## 2 Preliminaries

We use $\mathbb{P}(\cdot)$ to denote probability mass, $\mathbb{E}[\cdot]$ to denote expectation and $\mathbb{1}\{\cdot\}$ to denote an indicator function. For a positive integer $M$, we use $[M]$ as a short for $\{1, \ldots, M\}$. For a vector of $M$ elements $(x_1, \ldots, x_M)$, a *position $j$* is a number in $[M]$ used to identify the element $x_j$. A vector is said to be *diagonal* if its elements in all positions are equal. Suppose $f$ is a function on a domain $\mathcal{X}$ and $\boldsymbol{x}$ is a vector of $M$ elements in $\mathcal{X}$, we use the symbol $f(\boldsymbol{x})$ to denote the vector $(f(x_1), \ldots, f(x_M))$.

**Classifiers.** Let $\mathcal{X}$ denote an instance space and $\mathcal{Y}$ denote an output space with $|\mathcal{Y}| = c$. Let also $\mathcal{D}$ be the joint distribution of two random variables $(X, Y) \in \mathcal{X} \times \mathcal{Y}$ and $\mathcal{D}_X$ be the marginal of $X$. Throughout the text, we use $\boldsymbol{x}$ for $(x_1, \ldots, x_M)$ and $\boldsymbol{y}$ for the corresponding vector of gold labels. We consider *scoring functions* of the form $f : \mathcal{X} \to \Delta_c$, where $\Delta_c$ is the space of probability distributions on $\mathcal{Y}$ (e.g., $f$ outputs the softmax probabilities of a neural network). We use $f^j(x)$ to denote the $j$-th output of $f(x)$. A scoring function $f$ induces a *classifier* whose *prediction* on $x$ $[f](x)$ is defined by $[f](x) := \mathrm{argmax}_{j \in [c]} f^j(x)$. We use $\mathcal{F}$ and $[\mathcal{F}]$ to denote the space of scoring functions and the space of classifiers induced by $\mathcal{F}$, respectively. We use the VC-dimension, the Natarajan dimension and Rademacher complexities to characterize the complexity of $\mathcal{F}$ [40].

**Loss functions.** We define the problem of *learning* a scoring function using a fixed set of training samples as the one of *choosing* the scoring function from $\mathcal{F}$ that better *fits* the training samples. We give semantics to the term *fits* by using the *risk* $\mathcal{R}(f; \ell)$ of $f$ subject to a sampling distribution and a given *loss function* $\ell$: the lower $\mathcal{R}(f; \ell)$ becomes, the better $f$ fits the data. When there exists an $f^* \in \mathcal{F}$, such that $\mathcal{R}(f^*; \ell) = 0$, we say that the space $\mathcal{F}$ is *realizable* under $\ell$. In *supervised learning*, we are provided with samples of the form $(x, y)$ drawn from $\mathcal{D}$. We use $\widehat{\mathcal{R}}$ to denote the *empirical risk*, i.e., the average risk computed over the training samples only. The *risk*[2] of $f$ subject to a loss function $\ell : \mathcal{Y} \times \mathcal{Y} \to \mathbb{R}^+$ is given by $\mathcal{R}(f; \ell) := \mathbb{E}_{(X,Y) \sim \mathcal{D}}[\ell([f](X), Y)]$.

A commonly used loss function is the *zero-one* loss defined by $\ell^{01}(y, y') := \mathbb{1}\{y' \neq y\}$ for any pair $y, y' \in \mathcal{Y}$. We use $\mathcal{R}^{01}(f)$ as a short for $\mathcal{R}(f; \ell^{01})$ and refer to it as the *zero-one risk of* $f$. We will also consider *Semantic Loss* (SL) [56], which has been adopted to train classifiers subject to Boolean formulas [30, 43, 24]. Let $\varphi$ be a Boolean formula where each variable in $\varphi$ is associated with a class from $\mathcal{Y}$. Then, the SL of $\varphi$ subject to $f(x)$ is the negative logarithm of the *weighted model counting* (WMC) [7] of $\varphi$ under $f(x)$, namely $\mathrm{SL}(\varphi, f(x)) := -\log(\mathrm{WMC}(\varphi, f(x)))$, where $\mathrm{WMC}(\varphi, f(x))$ takes values in $(0, 1)$ and denotes the probability that $\varphi$ is logically satisfied when its variables become true with probabilities specified by $f(x)$.

## 3 Learning a Single Classifier Under a Known Transition

We begin with the setting where the goal is to learn a single classifier $f \in \mathcal{F}$ under a known transition $\sigma$. Firstly, we show ERM-learnability (Theorem 1). Then, inspired by neuro-symbolic learning, we provide a Rademacher-style error bound with a top-$k$ surrogate loss based on WMC (Theorem 2). Below, we formally introduce the learning setting.

**Problem setting.** Let $\sigma : \mathcal{Y}^M \to \mathcal{S}$ be a transition function, where $\mathcal{S} = \{1, \ldots, d\}$ with $|\mathcal{S}| = d \geq 1$. Let also $\mathcal{T}_\mathsf{P}$ be a set of $m_\mathsf{P}$ *partially labeled* samples of the form $(\boldsymbol{x}, s) = (x_1, \ldots, x_M, s)$. Each training sample is formed by, firstly drawing $M$ i.i.d. samples $(x_i, y_i)$ from $\mathcal{D}$ and then setting $s = \sigma(y_1, \ldots, y_M)$. We use $\mathcal{D}_\mathsf{P}$ to denote the distribution followed by the training samples $(\boldsymbol{x}, s)$. In analogy to the zero-one classification loss $\ell^{01}$, we define the *zero-one partial loss* subject to $\sigma$ as $\ell^{01}_\sigma(\boldsymbol{y}, s) := \mathbb{1}\{\sigma(\boldsymbol{y}) \neq s\}$, for any $\boldsymbol{y} \in \mathcal{Y}^M$ and $s \in \mathcal{S}$. The learner aims to find the classifier $f$ with the minimal *zero-one risk* $\mathcal{R}^{01}(f)$. As the gold labels are hidden, the learner uses the dataset $\mathcal{T}_\mathsf{P}$ to estimate and minimize the *zero-one partial risk* of $f$ subject to $\sigma$ defined as $\mathcal{R}^{01}_\mathsf{P}(f; \sigma) := \mathbb{E}_{(X_1, \ldots, X_M, S) \sim \mathcal{D}_\mathsf{P}}[\ell^{01}_\sigma(([f](X_1), \ldots, [f](X_M)), S)]$.

We demonstrate these notions via Example 1. There, $f$ is an MNIST classifier, $\mathcal{X}$ is the space of MNIST images and $\mathcal{Y} = \{0, \ldots, 9\}$. The training samples are of the form $(x_1, x_2, s)$ where $s = \sigma(y_1, y_2) = y_1 + y_2$. The partial risk of $f$ is then the probability of predicting the wrong sum.

**Learnability.** A *partial learning algorithm* $\mathcal{A}$ takes a partially labeled dataset $\mathcal{T}_\mathsf{P}$ with $|\mathcal{T}_\mathsf{P}| = m_\mathsf{P}$ as input and outputs a function $\mathcal{A}(\mathcal{T}_\mathsf{P}) \in \mathcal{F}$. Similar to standard PAC learning (see, for example, [40]), we say a multi-instance problem is *learnable*, if there exists a partial learning algorithm $\mathcal{A}$, such that for any data distribution $\mathcal{D}$ over $\mathcal{X} \times \mathcal{Y}$ and any $\delta, \epsilon \in (0, 1)$, there is an integer $m_{\epsilon, \delta}$, such that $m_\mathsf{P} \geq m_{\epsilon, \delta}$ implies $\mathcal{R}^{01}(\mathcal{A}(\mathcal{T}_\mathsf{P})) \leq \epsilon$ with probability at least $1 - \delta$ with respect to $\mathcal{D}_\mathsf{P}$. We consider $\mathcal{A}$ to be an Empirical Risk Minimizer (ERM) algorithm that finds the classifier $f \in \mathcal{F}$ which minimizes the empirical zero-one partial risk $\widehat{\mathcal{R}}^{01}_\mathsf{P}(f; \sigma; \mathcal{T}_\mathsf{P}) := \sum_{(\boldsymbol{x}, s) \in \mathcal{T}_\mathsf{P}} \ell^{01}_\sigma(([f](x_1), \ldots, [f](x_M)), s) / m_\mathsf{P}$.

---

[2]As stated above, the risk is defined subject to a sampling distribution. For brevity, we will omit that distribution when defining the risk and fix the sampling distribution in each case.

### 3.1  ERM Learnability

To prove learnability under the partial zero-one loss, we aim to bound $\mathcal{R}^{01}(f)$ with the partial risk $\mathcal{R}_{\mathsf{P}}^{01}(f;\sigma)$. Firstly, we propose a sufficient and necessary condition called $M$-*unambiguity*.

Recall that learnability requires handling all possible distributions of $(\boldsymbol{x}, \boldsymbol{y})$. Therefore, to propose a *necessary* condition to guarantee learnability, we consider a special case: the one in which $\mathcal{D}$ concentrates its mass on a single item $x_0$, i.e., $\mathbb{P}_{\mathcal{D}_{\mathcal{X}}}(x_0) = 1$. In that case, the only sample observed during learning includes only the instance $x_0$. Now, let $l_0 \in \mathcal{Y}$ be the gold label of $x_0$. If $f$ mispredicts the class of $x_0$, i.e., $f(x_0) \neq l_0$, and this misprediction leads to the same observed partial label, i.e., $\sigma(f(x_0), \ldots, f(x_0)) = \sigma(l_0, \ldots, l_0)$, then the problem will be not learnable, as we will never be able to correct this error. Motivated by this special case, we propose the following condition:

**Definition 1** ($M$-unambiguity)**.** *Transition $\sigma$ is $M$-unambiguous if for any two diagonal label vectors $\boldsymbol{y}$ and $\boldsymbol{y}' \in \mathcal{Y}^M$ such that $\boldsymbol{y} \neq \boldsymbol{y}'$, we have that $\sigma(\boldsymbol{y}') \neq \sigma(\boldsymbol{y})$.*

Recall that a vector is diagonal if all its elements are equal. Below, we provide examples of transitions that satisfy (or not) the $M$-unambiguity condition.

**Example 2.** *Consider the following learning problems.*

- *(SUM-$M$): Consider a variant of Example 1 where $s$ is the sum of $M$ digits, i.e., $\sigma(y_1, \ldots, y_M) = \sum_{i=1}^{M} y_i$. The transition is $M$-unambigous, since $\sigma(y, \ldots, y) \neq \sigma(y', \ldots, y')$ holds for any $y \neq y'$.*

- *(PRODUCT-$M$): Consider a variant of Example 1 where $s$ is given by $\sigma^*(y_1, \ldots, y_M) = \prod_{i=1}^{M} y_i$. Then, $\sigma^*$ is $M$-unambiguous, since $\sigma^*(y, \ldots, y) = y^M \neq (y')^M$ holds for any $y \neq y'$.*

- *(XOR): Consider Boolean labels. We take the XOR of the two digits as the partial label, i.e., $\sigma^{XOR}(y_1, y_2) = \mathbb{1}\{y_1 \neq y_2\}$. Then $\sigma^{XOR}$ is not $M$-unambiguous, since $\sigma^{XOR}(1, 1) = \sigma^{XOR}(0, 0)$.*

Although $M$-unambiguity is proposed as a necessary condition under special types of data distributions, we now show that it is also *sufficient* for proving ERM-learnability. Firstly, we prove that the classification risk can be bounded by the partial risk.

**Lemma 1.** *If $\sigma$ is $M$-unambigous, then we have:*

$$\mathcal{R}^{01}(f) \leq \mathcal{O}(\mathcal{R}_{\mathsf{P}}^{01}(f;\sigma)^{1/M}) \quad as \quad \mathcal{R}_{\mathsf{P}}^{01}(f;\sigma) \to 0 \tag{1}$$

*Moreover, if $\sigma$ is not $M$-unambiguous, then learning from partial labels is arbitrarily difficult, in the sense that a classifier $f$ with partial risk $\mathcal{R}_{\mathsf{P}}^{01}(f;\sigma) = 0$ can have a risk of $\mathcal{R}^{01}(f) = 1$.*

To show learnability, we bound the partial risk with its empirical counterpart in the realizable case.

**Theorem 1** (ERM learnability under $M$-unambiguity)**.** *Suppose $\mathcal{F}$ is realizable under $\ell_{\mathsf{P}}^{01}$ and $[\mathcal{F}]$ has a finite Natarajan dimension $d_{[\mathcal{F}]}$. Then for any $\epsilon, \delta \in (0, 1)$, there exists a universal[3] constant $C_0 > 0$, such that with probability at least $1 - \delta$, the empirical partial risk minimizer with $\widehat{\mathcal{R}}_{\mathsf{P}}^{01}(f;\sigma;\mathcal{T}_{\mathsf{P}}) = 0$ has a classification risk $\mathcal{R}^{01}(f) < \epsilon$, if*

$$m_{\mathsf{P}} \geq C_0 \frac{c^{2M-2}}{\epsilon^M} \left( d_{[\mathcal{F}]} \log(6cMd_{[\mathcal{F}]}) \log\left(\frac{c^{2M-2}}{\epsilon^M}\right) + \log\left(\frac{1}{\delta}\right) \right) \tag{2}$$

Beyond Lemma 1, Theorem 1 builds upon several non-trivial intermediate results: (i) A bound of the VC dimension for the partial label predictor (Lemma 3 in the Appendix) and (ii) Construction of counter-examples for arguing the necessity of $M$-unambiguity (see the proof of Theorem 1).

**Toward a faster convergence rate.** Theorem 1 presents a rather slow convergence rate of $(c^2/\epsilon)^M$. In the following, we show that a better convergence rate is achievable by forcing stricter ambiguity conditions. In the following, we introduce the concept of *1-unambiguity*, which requires the transition to be sensitive to 1-position perturbations:

**Definition 2** (1-unambiguity)**.** *Transition $\sigma$ is 1-unambiguous, if there exists an index $1 \leq i \leq M$, such that flipping the $i$-th label of any label vector $\boldsymbol{y} \in \mathcal{Y}^M$, results in a vector $\boldsymbol{y}'$ with $\sigma(\boldsymbol{y}') \neq \sigma(\boldsymbol{y})$.*

In the following, we provide examples of transitions that satisfy (or not) the 1-unambiguity condition.

---

[3] A constant is universal if it that does not depend on the parameters of the learning problem (e.g., $M$ or $c$).

**Example 3.** *Let us continue with Example 2. The transition $\sigma$ from SUM-M is 1-unambiguous since $\sigma(\boldsymbol{y}) = \sum_{i=1}^{M} y_i$ always change when replacing any of the labels $y_i$. However, the transition $\sigma^*$ from PRODUCT-M is not 1-unambiguous, since for the label vector $\boldsymbol{y} = (0, 1, 1)$ (with product zero), the flipped vector $\boldsymbol{y}' = (0, 1, 2)$ also leads to product zero.*

We show that a better convergence rate can be achieved if $\sigma$ is both 1- and $M$-unambigous.

**Proposition 1** (ERM learnability under 1- and $M$-unambiguity). *If $\sigma$ is both 1- and $M$-unambigous, then we have:*

$$\mathcal{R}^{01}(f) \leq \mathcal{O}(\mathcal{R}_{\mathsf{P}}^{01}(f; \sigma)) \quad as \quad \mathcal{R}_{\mathsf{P}}^{01}(f; \sigma) \to 0 \tag{3}$$

*Furthermore, if $\mathcal{F}$ is realizable under $\ell_{\mathsf{P}}^{01}$ and $[\mathcal{F}]$ has a finite Natarajan dimension $d_{[\mathcal{F}]}$, then for any $\delta \in (0, 1)$ and $\epsilon \in (0, 1)$ that is sufficiently close to 0, there exists a universal constant $C_1$, such that with probability at least $1 - \delta$, the empirical partial risk minimizer with $\widehat{\mathcal{R}}_{\mathsf{P}}^{01}(f; \sigma) = 0$ has a classification risk $\mathcal{R}^{01}(f) < \epsilon$, if*

$$m_{\mathsf{P}} \geq C_1 \frac{1}{\epsilon} \left( d_{[\mathcal{F}]} \log(6cMd_{[\mathcal{F}]}) \log \left( \frac{2}{\epsilon} \right) + \log \left( \frac{1}{\delta} \right) \right) \tag{4}$$

**Remark 1.** The supervision power of a partial label $s$ depends on the "instability" of $\sigma$: if $\sigma$ returns different results under certain perturbations in $I$ positions for an $I \in [M]$, then the classification risk is bounded by $\mathcal{R}^{01}(f) \leq \mathcal{O}(\mathcal{R}_{\mathsf{P}}^{01}(f; \sigma)^{1/I})$. To formalize this intuition, in the appendix, we provide a generalized definition of both 1- and $M$-unambiguity, called $I$-unambiguity, and show learnability with intermediate convergence rates. Notice that $M$-unambiguity is closely related to the small ambiguity degree condition in the PLL literature [28]. Appendix E.3 provides a relevant discussion and an extension of our results for randomized transitions.

**Remark 2.** We use the concentrated distribution described at the beginning of Section 3.1 as a pivot to design a necessary and sufficient learnability condition. However, the results in Lemma 1, Theorem 1 and Proposition 1 do not apply only to this distribution, but to any distribution $\mathcal{D}_{\mathsf{P}}$ of partial data. The same applies to all learnability results that follow.

### 3.2 Error Bounds with the Top-$k$ Semantic Loss

We now study the error bounds with the semantic loss (see Section 2 and Appendix A) under top-$k$ approximations (Theorem 2). We are interested in this loss for two reasons. Firstly, risk minimization under the zero-one loss is intractable even for linear classifiers [17] and hence in practice, learning is performed by minimizing differentiable surrogate losses such as the semantic loss. Secondly, computing a surrogate loss typically introduces another source of intractability: that of enumerating the entries of $\sigma$ [24, 43, 31]. For example, in SUM-$M$, we have to enumerate $10^M$ digit combinations to populate $\sigma$, which is not scalable.

A popular way to reduce the second source of inefficiency is to consider only the $k$ most probable label combinations during training, where the probability of a label vector $\boldsymbol{y}$ given $\boldsymbol{x}$ and classifier $f$ is defined by $P_{f(\boldsymbol{x})}(\boldsymbol{y}) := \prod_{i=1}^{M} f^{y_i}(x_i)$. Then, training can proceed by taking the semantic loss over the top-$k$ label vectors [24]. The above is possible, as the top-$k$ label vectors can be equivalently viewed as a formula that is true iff one or more of the top-$k$ label vectors is the gold one.

**Example 4.** *Let us return back to Example 1. Instead of considering all three combinations $(2, 0)$, $(1, 1)$ and $(0, 2)$ assuming a target $s = 2$, we can consider only the first and the last combination, if the probabilities (predicted by $f$) of the two images being 1 are way lower than those of being 0 or 2. Then, the top-2 label vectors $(2, 0)$, $(0, 2)$ can be seen as the formula $(A_{1,2} \wedge A_{2,0}) \vee (A_{1,0} \wedge A_{2,2})$, where $A_{i,j}$ is a Boolean variable that is true iff the $i$-th input digit is assigned label $j$.*

We use $\bigvee_{i=1}^{k} \boldsymbol{y}_i$ to denote the Boolean formula computed out of the $k$ label vectors $\boldsymbol{y}_1, \ldots, \boldsymbol{y}_k$, where each $\boldsymbol{y}_i$ is the conjunction of Boolean variables of the form $A_{i,y}$, as in Example 4. Given a softmax score $f(\boldsymbol{x})$, variable $A_{i,y}$ is assigned probability $f^y(x_i)$. We now define the *top-k partial loss*:

**Definition 3** (Top-$k$ partial loss). *The top-$k$ partial loss for an integer $k \geq 1$ subject to a classifier $f$, transition $\sigma$ and sample $(\boldsymbol{x}, s)$ is defined as*

$$\ell_{\sigma}^{k}(f(\boldsymbol{x}), s) := \mathrm{SL} \left( \bigvee_{i=1}^{k} \boldsymbol{y}^{(i)}, f(\boldsymbol{x}) \right) \tag{5}$$

*where $\boldsymbol{y}^{(1)}, \ldots, \boldsymbol{y}^{(k)} \in \mathcal{Y}^M$ are the top-$k$ maximizers of $P_{f(\boldsymbol{x})}$ in the preimage of $s$, i.e., $\{\boldsymbol{y} \in \mathcal{Y}^M : \sigma(\boldsymbol{y}) = s\}$. The* top-$k$ partial classification risk *of $f$ subject to $\sigma$ is then given by*

$$\mathcal{R}_{\mathsf{P}}^k(f; \sigma) := \mathbb{E}_{(X_1, \ldots, X_M, S) \sim \mathcal{D}_{\mathsf{P}}}[\ell_{\sigma}^k(f(X_1, \ldots, X_M), S)] \tag{6}$$

**Remark.** The WMC is upper bounded by the sum of probabilities, so we can approximate the SL as $\ell_{\sigma}^k(f(\boldsymbol{x}), s) \geq -\log(\sum_{i=1}^k P_{f(\boldsymbol{x})}(\boldsymbol{y}^{(i)}))$. Furthermore, the special case where $k = 1$ reduces to the infimum loss [5] and minimal loss [29] in the PLL literature as we show in Appendix B.2.

Let $\mathfrak{R}_m(\mathcal{F})$ denote the Rademacher complexity [4, 10] of $\mathcal{F}$ with $m$ samples as defined in Appendix A. We are now ready to bound the zero-one classification risk with the empirical top-$k$ partial risk.

**Theorem 2** (Error bound under unambiguity). *Let an integer $k \geq 1$ and $\delta \in (0, 1)$. If $\sigma$ is both 1- and $M$-unambiguous, then with probability at least $1 - \delta$, we have:*

$$\mathcal{R}^{01}(f) \leq \Phi \left( (k+1) \left( \widehat{\mathcal{R}}_{\mathsf{P}}^k(f; \sigma; \mathcal{T}_{\mathsf{P}}) + 2\sqrt{k} M^{3/2} \mathfrak{R}_{M m_{\mathsf{P}}}(\mathcal{F}) + \sqrt{\frac{\log(1/\delta)}{2 m_{\mathsf{P}}}} \right) \right) \tag{7}$$

*where $\widehat{\mathcal{R}}_{\mathsf{P}}^k(f; \sigma; \mathcal{T}_{\mathsf{P}}) = \sum_{(\boldsymbol{x}, s) \in \mathcal{T}_{\mathsf{P}}} \ell_{\sigma}^k(f(\boldsymbol{x}), s)/m_{\mathsf{P}}$ is the empirical counterpart of (6) and $\Phi$ is an increasing function that satisfies $\lim_{t \to 0} \Phi(t)/t = 1$.*

*Proof sketch.* Theorem 2 builds upon several results. Firstly, we derived an inequality that bounds the top-$k$ loss with the zero-one loss (Lemma 5), which requires the construction of an intermediate $\ell^1$ loss (Definition 11). Secondly, we show the Lipschtness of the semantic loss (Lemma 7). This result is further combined with a contraction lemma that is proposed in [10] (Lemma 6) to bound the Rademacher complexity of the model. □

**Remark.** Similarly to Theorem 1 and Proposition 1, Theorem 2 suggests that the learning difficulty increases as $M$ increases. This is intuitive, since it gets harder to disambiguate the gold labels when $M$ increases, e.g., for 2-SUM, there are $10^2$ possible label vectors need to be considered, while for 4-SUM, there are $10^4$ ones. Our bound also suggests a tradeoff with the choice of $k$: a larger $k$ decreases the risk, but tends to increase the complexity term.

## 4 Learning Multiple Classifiers Under a Known Transition

In this section, we extend the learning problem from Section 3 to jointly learn $n \geq 1$ different classifiers. Similarly to Section 3, we aim to propose minimal assumptions for the data distribution to show both results on ERM-learnability (Theorem 3) and a Rademacher-style error bound with the top-$k$ loss (Theorem 4). We formally define the learning setting below.

**Problem setting.** We aim to learn $n \geq 1$ classifiers, each of which maps instances from a space $\mathcal{X}_i$ to the corresponding label space $\mathcal{Y}_i$ with $|\mathcal{Y}_i| = c_i$. For each $i \in [n]$, we define a scoring space $\mathcal{F}_i$, which contains mappings of the form $f_i : \mathcal{X}_i \to \Delta_{c_i}$. Each $f_i$ is used to classify $M_i \geq 1$ instances $(x_{i1}, \ldots, x_{iM_i}) = \boldsymbol{x}_i \in \mathcal{X}_i^{M_i}$ with (hidden) gold labels $\boldsymbol{y}_i = (y_{i1}, \ldots, y_{iM_i})$. We denote the vector of scoring functions as $\boldsymbol{f} = (f_1, \ldots, f_n)$. Let $\mathcal{D}_i$ be the joint distribution over elements from $\mathcal{X}_i \times \mathcal{Y}_i$. Each training sample given to the learner $(\boldsymbol{x}_1, \ldots, \boldsymbol{x}_n, s)$ is formed by (i) drawing $M_i$ i.i.d. samples $(\boldsymbol{x}_i, \boldsymbol{y}_i)$ from $\mathcal{D}_i$, for each $i \in [n]$ and then (ii) obtaining its partial label as $s = \sigma(\boldsymbol{y}_1, \ldots, \boldsymbol{y}_n)$. We override the notation $\mathcal{D}_{\mathsf{P}}$ to denote the distribution of the training samples in this learning setting. The partially labeled dataset $\mathcal{T}_{\mathsf{P}}$ then contains $m_{\mathsf{P}}$ i.i.d. samples drawn from $\mathcal{D}_{\mathsf{P}}$.

Similarly to Section 3, the learner aims to find the classifiers $f_1, \ldots, f_n$ with the minimal *zero-one risk* defined as $\mathcal{R}^{01}(\boldsymbol{f}) := \sum_{i=1}^n \mathcal{R}^{01}(f_i)$. As the gold labels are hidden, the learner uses the dataset $\mathcal{T}_{\mathsf{P}}$ to estimate and minimize the *zero-one partial risk* of $f_1, \ldots, f_n$, which is defined as $\mathcal{R}_{\mathsf{P}}^{01}(f_1, \ldots, f_n; \sigma) := \mathbb{E}_{(\mathbf{X}_1, \ldots, \mathbf{X}_n, S) \sim \mathcal{D}_{\mathsf{P}}}[\ell_{\sigma}^{01}(([f_1](\mathbf{X}_1), \ldots, [f_n](\mathbf{X}_n)), S)]$, where $[f_i](\mathbf{X}_i)$ is a short for $([f_i](X_{i1}), \ldots, [f_i](X_{iM_i}))$, for $1 \leq i \leq M_i$.

**Example 5** (Learning binary operators). *Consider a setting where we aim to train a classifier $f_1$ for recognizing MNIST digits and a classifier for recognizing images of addition and multiplication taken from some space $\mathcal{X}_2$. The training samples are of the form $(x_{11}, x_{12}, x_{21}, s)$, where $x_{11}$ and $x_{12}$ are non-zero MNIST digits with $\mathcal{Y}_1 = \{1, \ldots, 9\}$, $x_{21}$ is an image of an operator with $\mathcal{Y}_2 = \{+, \times\}$, and $s$ is the result of applying the operator in $x_2$ on the digits in $x_{11}$ and $x_{12}$.*

## 4.1 ERM Learnability

Similarly to Section 3, to propose a necessary condition for learnability, we consider the most challenging distribution where each $\mathcal{D}_i$ is concentrated on a single instance $x_i^* \in \mathcal{X}_i$, for $1 \leq i \leq n$. Then, the only sample that will be observed during learning is $(\boldsymbol{x}_1^*, \ldots, \boldsymbol{x}_n^*)$ where $\boldsymbol{x}_i = (x_i^*, \ldots, x_i^*)$, for $1 \leq i \leq n$. Assuming that $l_i^*$ is the gold label of $x_i^*$, detecting a classification error is possible only if a misclassification of any $x_i^*$, i.e., $[f_i](x_i^*) \neq l_i^*$, will lead to a prediction vector $\boldsymbol{y}'$ with a different image under $\sigma$ from the gold label $\boldsymbol{y}$, i.e., $\sigma(\boldsymbol{y}') \neq \sigma(\boldsymbol{y})$. The above intuition is captured via the notion of *multi-unambiguity*, which generalizes the notion of $M$-unambiguity. Below, we use $\boldsymbol{y}_i^{M_i}$, for $i \in [n]$, to denote the $M_i$-ary diagonal vector including the $y_i$ element only.

**Definition 4** (Multi-unambiguity). *Transition $\sigma$ is* multi-unambiguous *if for any vector $\boldsymbol{y} = (\boldsymbol{y}_1^{M_1}, \ldots, \boldsymbol{y}_n^{M_n})$, and any position $i \in [n]$, such that the vector $\boldsymbol{y}'$ that results after flipping the labels in $\boldsymbol{y}_i^{M_i}$ to some diagonal vector $(\boldsymbol{y}_i')^{M_i} \neq \boldsymbol{y}_i^{M_i}$, has a different image under $\sigma$, i.e., $\sigma(\boldsymbol{y}) \neq \sigma(\boldsymbol{y}')$.*

Multi-unambiguity reduces to $M$-unambiguity for $n = 1$. An example of multi-unambiguity is below.

**Example 6** (Learning binary operator, cont'd). *In Example 5, the multi-unambiguity condition is violated since $2 + 2 = 2 \times 2$. Namely, if a distribution assigns all its weight to the digit $2$, then it is difficult for the model to distinguish the two operators. On the other hand, if instead, the label space is $\mathcal{Y}_1 = \{3, \ldots, 9\}$, then the multi-unambiguity condition is satisfied.*

**Bounded risk assumption.** Differently from Section 3, multi-unambiguity alone is not sufficient to ensure learnability. To see this, consider a variant of SUM2 where the digits are classified by two independent classifiers, $f_1$ and $f_2$. If the distributions of the first and the second image are concentrated on 1 and 7 respectively, but $f_1(x_1) = 7$ and $f_2(x_2) = 1$, then these errors cannot be detected. Therefore, we assume there is a constant $R < 1$, such that $\mathcal{R}^{01}(f_i) \leq R$ holds for any $i \in [n]$ and any $f \in \mathcal{F}_i$. We refer to such $f_i$'s as *zero-one risk $R$-bounded*. This assumption is mild since it only requires the classifiers to be slightly better than being totally wrong. Also, when a small directly labeled dataset is available, one can bound the classification risks away from 1 with high probability in a uniform manner using standard learning theory. See Appendix C for more details. Now, we are ready to state the ERM-learnability result assuming $\prod_{i=1}^n \mathcal{F}_i$ is realizable under $\ell_{\mathsf{P}}^{01}$.

**Theorem 3** (ERM learnability under multi-unambiguity). *Assume that there is a constant $R < 1$, such that for each $i \in [n]$, each $f \in \mathcal{F}_i$ is zero-one risk $R$-bounded. Assume also that there exist positive integers $M^*$ and $c^*$, such that $M_i \leq M^*$ and $c_i \leq c_0$ hold for any $i \in [n]$. Then, if $\sigma$ is multi-unambiguous, we have:*

$$\mathcal{R}^{01}(\boldsymbol{f}) \leq \mathcal{O}((\mathcal{R}_{\mathsf{P}}^{01}(\boldsymbol{f}; \sigma))^{1/M^*}) \quad as \quad \mathcal{R}_{\mathsf{P}}^{01}(\boldsymbol{f}; \sigma) \to 0 \tag{8}$$

*Furthermore, for any $\epsilon, \delta \in (0, 1)$, there is a universal constant $C_3$, such that with probability at least $1 - \delta$, the empirical partial risk minimizer with $\widehat{\mathcal{R}}_{\mathsf{P}}^{01}(f; \sigma) = 0$ has a classification risk $\mathcal{R}^{01}(f) < \epsilon$ if*

$$m_{\mathsf{P}} \geq C_3 \frac{nc_0^{2M^*-2}}{\epsilon^{M^*}(1-R)^M} \left( \sum_{i=1}^n d_{[\mathcal{F}_i]} \log(nc_i M_i d_{[\mathcal{F}_i]}) \log\left( \frac{nc_0^{2M^*-2}}{\epsilon^{M^*}(1-R)^M} \right) + \log\left( \frac{1}{\delta} \right) \right) \tag{9}$$

**Remark.** Theorem 3 suggests that the rate of convergence is controlled by $M^*$. A smaller $M^*$ leads to a more strict unambiguity condition. For example, the case $M^* = 1$ requires $\sigma$ to be unstable to *any* 1-position perturbations, while the case $M^* = M$ reduces to $M$-unambiguity. This observation confirms the intuition that the supervision power of partial labels depends on the "instability" of $\sigma$.

The top-$k$ partial loss $\ell_\sigma^k(\boldsymbol{f}(\boldsymbol{x}), s)$ for multiple classifiers subject to $\boldsymbol{f} \in \mathcal{F}^n$, transition $\sigma$ and $s \in \mathcal{S}$ straightforwardly extends the top-$k$ partial loss for the single classifier case, see Definition 3. Similarly, the *(empirical) top-$k$ partial classification risk* of $\boldsymbol{f}$ subject to $\sigma$ straightforwardly extends the one from Section 3.2. Below, we provide a Rademacher-style error bound with the top-$k$ loss.

**Theorem 4** (Error bound under multi-unambiguity with multiple classifiers). *Suppose $\sigma$ is multi-unambiguous and each $f_i$ is zero-one risk $R$-bounded, for $R \in (0, 1)$. Then, for any integer $k \geq 1$ and any $\delta \in (0, 1)$, with probability at least $1 - \delta$, we have:*

$$\mathcal{R}^{01}(\boldsymbol{f}) \leq \left( \frac{nc_0^{2M^*-2}(k+1)}{(1-R)^M} \left( \widehat{\mathcal{R}}_{\mathsf{P}}^k(\boldsymbol{f}; \sigma; \mathcal{T}_{\mathsf{P}}) + \sqrt{kM} \sum_{i=1}^n M_i \mathfrak{R}_{m_{\mathsf{P}} M_i}(\mathcal{F}_i) + \sqrt{\frac{\log(1/\delta)}{2m_{\mathsf{P}}}} \right) \right)^{1/M^*}$$

# 5 Learning Under an Unknown Transition

We now explore another direction by dropping the assumption that $\sigma$ is known. Instead, we assume the learner has access to a *transition space* $\mathcal{G}$ that contains mappings of the form $\mathcal{Y}^M \to \mathcal{S}$ including the "true" transition $\sigma$. Notice that learning now becomes particularly challenging, since $\mathcal{G}$ can be expressive enough to lead to correct predictions for $s$ from many wrong classifications.

**Problem setting.** To illustrate the core idea, we consider a simple setting where we only learn a single scoring function $f \in \mathcal{F}$ as in Section 3. However, $\sigma$ is an unknown mapping in $\mathcal{G}$. Given partially labeled samples, the learner aims to minimize the *zero-one partial risk of $f$ subject to $\mathcal{G}$* $\mathcal{R}_\mathsf{P}^{01}(f; \mathcal{G})$ that is defined as the *minimal* partial risk to predict $s$ with a candidate transition in $\mathcal{G}$, i.e., $\mathcal{R}_\mathsf{P}^{01}(f; \mathcal{G}) := \inf_{\sigma^* \in \mathcal{G}} \mathcal{R}_\mathsf{P}^{01}(f; \sigma^*)$. The risk is empirically estimated with a partially labeled dataset $\mathcal{T}_\mathsf{P}$ as $\inf_{\sigma^* \in \mathcal{G}} \widehat{\mathcal{R}}_\mathsf{P}^{01}(f; \sigma^*; \mathcal{T}_\mathsf{P})$. We demonstrate this learning setting with the following example:

**Example 7.** *Let $\mathcal{G} = \{(y_1, y_2) \mapsto \alpha y_1 + \beta y_2 | (\alpha, \beta) \in \mathbb{R}^2 - \{(0,0)\}\}$, namely all the weighted sums with at least one non-zero weight. We aim to learn an MNIST classifier $f$, given training samples of the form $(x_1, x_2, s)$, where the $x_i$'s are MNIST digits and $s$ is the result of applying some $\sigma$ from $\mathcal{G}$ on $(y_1, y_2)$, where $y_i$ is the prediction of $f$ on $x_i$. The gold $\sigma$, i.e., the exact $\alpha$ and $\beta$, is unknown.*

## 5.1 ERM Learnability

Similarly to the known transition case, learnability requires us to learn under any data distribution $\mathcal{D}$. Slightly differently from Sections 3 and 4, we start with the adversarial distribution where the mass in $\mathcal{D}$ is concentrated to a single label, i.e., all instances $x_i$ have the same gold label $l_i$, and hence, the training samples are all associated with the gold label vector $\boldsymbol{y} = (l_i, \ldots, l_i)$. Suppose there is a certain probability that $f$ misclassifies $l_i$ as $l_j$. Then, the predicted label vectors will be in $\{l_i, l_j\}^M$. In this case, it will be impossible to detect the classification errors if there is a candidate transition $\sigma' \in \mathcal{G}$ which maps all the vectors in $\{l_i, l_j\}^M$ to the gold observation of $s$. Since the actual $\sigma$ is unknown, we require this does not happen for any pair of $\sigma, \sigma' \in \mathcal{G}$ to ensure learnability. Below, we formalize our learnability condition:

**Definition 5** (Unambiguous transition space). *Transition space $\mathcal{G}$ is* unambiguous*, if for each $\sigma' \in \mathcal{G}$, each diagonal label vector $\boldsymbol{y} = (l_i, \ldots, l_i)$, where $l_i \in \mathcal{Y}$, and each $l_j \neq l_i$, where $l_j \in \mathcal{Y}$, there exists a vector $\boldsymbol{y}' \in \{l_i, l_j\}^M$, such that $\sigma'(\boldsymbol{y}') \neq \sigma(\boldsymbol{y})$.*

In pratice, one can examine whether a transition space is unambiguous by checking if for each $\sigma' \in \mathcal{G}$ and any two different labels $l_i \neq l_j$, set $\{\sigma'(\boldsymbol{y}) | \boldsymbol{y} \in \{l_i, l_j\}^M\}$ is not a singleton. The above ensures that when given a fixed diagonal label vector, and when the classifier makes mistakes, the predicted partial labels are not unique, and hence cannot all agree with the ground truth label.

Class $\mathcal{G}$ in Example 7 is unambiguous: for each transition $(y_1, y_2) \mapsto \alpha' y_1 + \beta' y_2$ and each two labels $l' \neq l$, the set $\{\alpha' y_1 + \beta' y_2 | (y_1, y_2) \in \{l, l'\}^2\}$ is not a singleton, so there must exist a label vector with a different weighted sum that is different than the true sum. A counterexample is below:

**Example 8.** *Let $\mathcal{G} = \{(y_1, y_2) \mapsto w_1 y_1 + w_2 y_1^2 + w_3 y_2^2 + w_4 y_2 | w_i \neq 0 \ \forall i \in [4]\}$. Consider the true transition $\sigma : (y_1, y_2) \mapsto y_1 - y_1^2 + y_2 - y_2^2$ from $\mathcal{G}$ and a candidate transition $\sigma' : (y_1, y_2) \mapsto y_1 - y_1^2 - y_2 + y_2^2$. Then, $\mathcal{G}$ is not unambiguous, since $\sigma'(\boldsymbol{y}) = \sigma(\boldsymbol{y}) = 0$ for any $\boldsymbol{y} \in \{0, 1\}^2$. Namely, the partially labeled data are not informative enough to distinguish the label $0$ from $1$.*

**Bounded risk assumption.** Similarly to Section 4, this unambiguity condition is necessary but not sufficient to show learnability. To show learnability, we additionally assume there is a constant $r > 0$ such that $\mathbb{P}([f](X) = y | Y = y) \geq r$ for each $y \in \mathcal{Y}$ and $f \in \mathcal{F}$. We refer to such an $f$ as *$r$-bounded*. This assumption is slightly stronger than that of Section 4 as it additionally requires the classifiers to be not fully wrong for *every* possible label. The main learnability result of this section is below:

**Theorem 5.** *If $\mathcal{G}$ is unambigous and any $f \in \mathcal{F}$ is $r$-bounded, then we have:*

$$\mathcal{R}^{01}(f) \leq O(\mathcal{R}_\mathsf{P}^{01}(f; \mathcal{G})^{1/M}) \quad \text{as} \quad \mathcal{R}_\mathsf{P}^{01}(f; \mathcal{G}) \to 0 \tag{10}$$

*Furthermore, suppose $[\mathcal{F}]$ has a finite Natarajan dimension $d_{[\mathcal{F}]}$ and the function class $\{(\boldsymbol{y}, s) \mapsto \mathbb{1}\{\sigma'(\boldsymbol{y}) \neq s\} | \sigma' \in \mathcal{G}\}$ has a finite VC-dimension $d_\mathcal{G}$. Then, for any $\epsilon, \delta \in (0, 1)$, there is a universal constant $C_4$ such that with probability at least $1 - \delta$, the empirical partial risk minimizer with*

$\widehat{\mathcal{R}}_{\mathsf{P}}^{01}(f;\sigma) = 0$ *has a classification risk* $\mathcal{R}^{01}(f) < \epsilon$, *if*

$$m_{\mathsf{P}} \geq C_4 \frac{c^{2M-2}}{r^M \epsilon^M} \left( \left( (d_{[\mathcal{F}]} + d_{\mathcal{G}}) \log(6M(d_{[\mathcal{F}]} + d_{\mathcal{G}})) + d_{[\mathcal{F}]} \log c \right) \log \left( \frac{c^{2M-2}}{r^M \epsilon^M} \right) + \log \left( \frac{1}{\delta} \right) \right)$$

## 6   Experiments

We further explore the learning scenario in Section 5 with empirical evaluations. We aim to learn an MNIST classifier using the weighted sum of 2, 3 and 4 MNIST digits and assuming that the weights are unknown, as in Example 7. The weighted sum function is very expressive as it can model Boolean formulas including conjunction and disjunction [36]. Notice that we do not aim to exhaustively assess the weak supervision literature ([24, 43, 59, 13]), but to validate the results of our theoretical analysis.

**Baselines.** We considered state-of-the-art neuro-symbolic frameworks, namely DeepProbLog (DLog) [30], DeepProbLog with approximations (DLog-A) [31], NeuroLog (NLog) [43], NeurASP (NASP) [59], ABL [13], ENT [27] and Scallop [24]. Only the target sum is used during training under those frameworks. DLog and NLog employ the semantic loss without any approximations (see Section 2). NLog($k$) denotes an NLog variant where only the top-$k$ predictions are considered during training, while Scallop($k$) denotes Scallop using the top-$k$ semantic loss, see Section 3.2. The approximations in DLog-A are different from the ones in NLog and Scallop; however, both DLog-A and NLog employ the semantic loss on the chosen subset of proofs. As the above frameworks learn classifiers under fixed theories only, we considered weights in $\{1, 2, 3, 4, 5\}$, and used an additional neural classifier to learn the unknown weights. Finally, we considered standard supervised learning (SL). To contrast learning under unknown to learning under known transitions, we also ran experiments assuming that the weights are known. Our results show that the classification accuracy exceeds the 98% even for $M = 10$. More details on this experiment are in the arXiv version of this paper [51].

**Results.** Tables 1 and 2 report the accuracy of the learned MNIST classifiers over ten runs. #0, #8 and #16 denote the number of directly labeled samples used for pretraining the MNIST classifiers: #0 means no pretraining; for #8 and #16, the accuracy of the pre-trained classifiers is 37% and 58%. Table 1 presents results for $M = 2$, while Table 2 presents results for $M \in \{2, 3, 4\}$ for Scallop and the two best performing frameworks in Table 1, DLog and NLog– NASP could not scale for $M \geq 3$. We used DLog-A for $M \in \{3, 4\}$, due to scalability reasons. Similarly, NLog without approximations does not scale for $M = 4$ (N/A indication). In Table 1, the number of weak samples is in parentheses, e.g., DLog(#4K). For ABL and ENT, we used 10K samples, as they rely on sampling, and used the hyperparameters suggested by the authors.

**Discussion.** The results confirm our theory: (i) we can learn classifiers under unknown transitions as per Theorem 5 (see the accuracy for DLog and NLog in Table 1); and (ii) learning gets harder when $M$ increases as per Eq. (5) (see the accuracy for different $M$'s in Table 2). For $M = 2$, weak supervision leads to better accuracy than SL even without pre-training and with fewer training samples: DLog(#4K) and NLog(#4K) lead to mean accuracy 96% and 97%, while SL(#4K) leads to mean accuracy 93%, see Table 1. This result suggests that the $r$-bounded assumption in Theorem 5 could be potentially relaxed. Another observation is that the approximations in DLog-A are less effective than the top-$k$ one of NLog. We leave the theoretical analysis of the latter approximation as part of future research. Table 2 also suggests that the top-$k$ semantic loss constitutes the most scalable and effective approach to training: in contrast to DLog-A and NLog, the accuracy of the digit classifier reaches 78% for $M = 4$, when $k = 4$ for Scallop.

Our empirical analysis shows that a key issue is *scalability*. This is a known problem in neuro-symbolic learning and it is partially because the relevant problems in logic are intractable. For instance, semantic loss requires computing the models of Boolean formulas, which is #P-complete. Reasoning over logical theories can be also intractable [2], if not undecidable. Scalability straightforwardly affects accuracy, e.g., NLog obtains better accuracy over DLog-A for $M = 3$ (see Table 2) due to its ability to explore the whole search space. It is also worth stressing that Scallop could not scale beyond $M = 4$ for $k > 1$. The above challenges bring up new research directions [44, 41]. Closing this discussion, it is worth exploring why ABL and ENT led to low accuracy[4].

---

[4]The authors from [43] also reported low accuracy for ABL for some scenarios.

Table 1: Classifier accuracy for WEIGHTED-SUM for $M = 2$. #0, #8 and #16 are # of directly labeled samples used for pre-training. Inside the parentheses are the # of (weak) training samples.

|  | SL(#4K) | SL(#8K) | DLog(#4K) | NLog(#4K) | NASP(#4K) | ABL(#10K) | ENT(#10K) |
|---|---|---|---|---|---|---|---|
| #0 | $93\% \pm 0.01$ | $96\% \pm 0.008$ | $96\% \pm 0.004$ | $97\% \pm 0.05$ | $98\% \pm 0.01$ | $9\% \pm 0.05$ | $40\% \pm 15.2$ |
| #8 | $93\% \pm 0.01$ | $96\% \pm 0.008$ | $96\% \pm 0.005$ | $97\% \pm 0.005$ | $98\% \pm 0.01$ | $10\% \pm 0.01$ | $43\% \pm 17.11$ |
| #16 | $93\% \pm 0.01$ | $96\% \pm 0.008$ | $96\% \pm 0.001$ | $97\% \pm 0.01$ | $98\% \pm 0.01$ | $11\% \pm 0.1$ | $49\% \pm 16.4$ |

Table 2: Classifier accuracy for WEIGHTED-SUM for $M \in \{2, 3, 4\}$. #0 and #16 are # of directly labeled samples used for pre-training. Inside the parentheses are the # of (weak) training samples.

|  | $M = 2, m_{\mathsf{P}} = 4K$ | | $M = 3, m_{\mathsf{P}} = 4K$ | | $M = 4, m_{\mathsf{P}} = 10K$ | |
|---|---|---|---|---|---|---|
|  | #0 | #16 | #0 | #16 | #0 | #16 |
| DLog-A | $34\% \pm 0.1$ | $80\% \pm 0.06$ | $24\% \pm 0.09$ | $27\% \pm 0.13$ | $18\% \pm 0.02$ | $26\% \pm 0.25$ |
| NLog | $97\% \pm 0.05$ | $97\% \pm 0.01$ | $61\% \pm 0.07$ | $97\% \pm 0.01$ | N/A | N/A |
| NLog(5) | $48\% \pm 0.003$ | $87\% \pm 0.007$ | $27\% \pm 0.06$ | $84\% \pm 0.06$ | $29\% \pm 0.1$ | $46\% \pm 0.05$ |
| Scallop(1) | $97\% \pm 0.05$ | $97\% \pm 0.01$ | $21\% \pm 3.5$ | $97\% \pm 0.01$ | $36\% \pm 0.2$ | $54\% \pm 0.02$ |
| Scallop(4) | $97\% \pm 0.05$ | $97\% \pm 0.05$ | $53\% \pm 0.15$ | $97\% \pm 0.07$ | $51\% \pm 0.09$ | $78\% \pm 0.05$ |

# 7   Related Work

We provide an overview of the literature that closely relates to our work. A more comprehensive review of PLL, latent structural learning, and neuro-symbolic learning, is presented in Appendix F.

**Partial label learning.** Learnability of PLL is typically shown under the *small ambiguity* assumption [11, 28, 5]. While it is technically possible to cast our problem to standard PLL by viewing a target $s$ as a partial label for the hidden labels $\boldsymbol{y}$, the small ambiguity condition is violated in our case since a distracting label can occur because $\sigma$ is deterministic. Therefore, our proposed conditions relax and generalize the small ambiguity by allowing deterministic transitions, see Appendix E.3. Another line of work in weak supervision exploits the invertibility of *transition matrices* [46] to compute posterior class probabilities [8, 9, 61]. Our learnability condition, $M$-unambiguity, and the invertibility condition do not imply each other, see Appendix E.1 and E.2. There, we also show that small ambiguity and the invertibility condition do not imply each other, which might be of independent interest. Another related topic is *learning with label constraints* [21, 6, 49], where the label of each instance $x$ (possibly structured) in the dataset is constrained in a subset $C \subseteq \mathcal{Y}$. The difference is that the constraint mapping itself is known to the learner and hence can be encoded in the inference algorithm directly, for example, via the CCM [6].

**Neuro-symbolic and structured learning.** We were motivated by frameworks that employ logic for training neural models [30, 53, 13, 59, 43, 31, 24, 27, 23]. We prove learnability under (unknown) transitions that capture different languages and error bounds under the SL [56] and top-$k$ approximations. Notice that [58] also proves the consistency of a top-$k$ loss. However, they use a zero-one-style loss for standard supervised learning. Our work is closely related to [62], which studies a similar problem of weakly supervised learning with multiple instances. Our results are stronger in the sense that we propose sufficient and necessary conditions to recover the hidden labels, while [62] concerns the likelihood of the observed labels rather than the hidden ones. Our work is also relevant to *latent structural prediction* [42, 35, 34, 32]. To our knowledge, no formal learning guarantees have been given for that problem. Complementary to ours is the work in [19, 47, 33] that integrates combinatorial solvers into deep models.

# 8   Conclusions

We formulated multi-instance PLL and showed its connections with latent structural and neuro-symbolic learning. Our work exhibits a greater level of flexibility compared to (single-instance) PLL, as it allows for multiple instances and deterministic transitions. We introduced minimal assumptions that enabled us to establish ERM-learnability and derive error bounds with the top-$k$ semantics loss. Our findings suggest that the interaction of multiple instances in forming the observed labels can help relax the learnability assumptions in weakly supervised learning. For future work, we will further relax the learning conditions by assuming a certain degree of smoothness for the data distributions. Another direction is to explore the setting where the hidden labels are non-i.i.d. or structured.

## Acknowledgements

This work was partially supported by Contract FA8750-19-2-0201 with the US Defense Advanced Research Projects Agency (DARPA). Approved for Public Release, Distribution Unlimited. The views expressed are those of the authors and do not reflect the official policy or position of the Department of Defense or the U.S. Government.

This work was also partially sponsored by the Army Research Office and was accomplished under Grant Number W911NF-20-1-0080. The views and conclusions contained in this document are those of the authors and should not be interpreted as representing the official policies, either expressed or implied, of the Army Research Office or the U.S. Government. The U.S. Government is authorized to reproduce and distribute reprints for Government purposes notwithstanding any copyright notation herein.

This work was also supported by Contract FA8750-19-2-1004 with the US Defense Advanced Research Projects Agency (DARPA). Approved for Public Release, Distribution Unlimited. The views expressed are those of the authors and do not reflect the official policy or position of the Department of Defense or the U.S. Government.

This work was also partially funded by ONR Contract N00014-19-1-2620.

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

# Appendix

This appendix is organized as follows:

- In Appendix A, we provide detailed definitions for the loss functions and the complexity measures we use in our work.
- In Appendix B, C and D, we provide the proofs to all formal statements.
- In Appendix F, we provide a comprehensive review of the related work in PLL, latent structural learning, contrained learning, and neuro-symbolic learning.
- In Appendix E, we compare common distributional assumptions in the standard PLL literature with ours. In particular:
    1. In Appendix E.1, we provide a transition matrix [8, 46] formulation of multi-instance PLL and show that $M$-unambiguity does not imply transition matrix invertibility for multi-instance PLL and vice versa.
    2. In Appendix E.2, we show a result of independent interest: that transition matrix invertibility does not imply small ambiguity [28] and vice versa.
    3. In Appendix E.3, we show that $M$-unambiguity, the unambiguity notion proposed in Definition 1, is an extension of the small ambiguity degree [28], by considering a generalized setting where $\sigma$ is stochastic.
- In Appendix G, we provide implementation details of our experiments.

## A  Preliminaries

Table 3: The $2^4$ interpretations of $\{A_{1,2}, A_{2,0}, A_{1,0}, A_{2,2}\}$ and their corresponding truth probabilities subject to the $\omega$ and $\varphi$ from Example 9. Interpretations that are not models of $\varphi$ have zero probability.

| $A_{1,2}$ | $A_{2,0}$ | $A_{1,0}$ | $A_{2,2}$ | $P_\varphi(I, \omega)$ |
|---|---|---|---|---|
| $\bot$ | $\bot$ | $\bot$ | $\bot$ | $0$ |
| $\bot$ | $\bot$ | $\bot$ | $\top$ | $0$ |
| $\bot$ | $\bot$ | $\top$ | $\bot$ | $0$ |
| $\bot$ | $\bot$ | $\top$ | $\top$ | $(1 - \omega(A_{1,2})) \times (1 - \omega(A_{2,0})) \times \omega(A_{1,0}) \times \omega(A_{2,2})$ |
| $\bot$ | $\top$ | $\bot$ | $\bot$ | $0$ |
| $\bot$ | $\top$ | $\bot$ | $\top$ | $0$ |
| $\bot$ | $\top$ | $\top$ | $\bot$ | $0$ |
| $\bot$ | $\top$ | $\top$ | $\top$ | $(1 - \omega(A_{1,2})) \times \omega(A_{2,0}) \times \omega(A_{1,0}) \times \omega(A_{2,2})$ |
| $\top$ | $\bot$ | $\bot$ | $\bot$ | $0$ |
| $\top$ | $\bot$ | $\bot$ | $\top$ | $0$ |
| $\top$ | $\bot$ | $\top$ | $\bot$ | $0$ |
| $\top$ | $\bot$ | $\top$ | $\top$ | $\omega(A_{1,2}) \times (1 - \omega(A_{2,0})) \times \omega(A_{1,0}) \times \omega(A_{2,2})$ |
| $\top$ | $\top$ | $\bot$ | $\bot$ | $\omega(A_{1,2}) \times \omega(A_{2,0}) \times (1 - \omega(A_{1,0})) \times (1 - \omega(A_{2,2}))$ |
| $\top$ | $\top$ | $\bot$ | $\top$ | $\omega(A_{1,2}) \times \omega(A_{2,0}) \times (1 - \omega(A_{1,0})) \times \omega(A_{2,2})$ |
| $\top$ | $\top$ | $\top$ | $\bot$ | $\omega(A_{1,2}) \times \omega(A_{2,0}) \times \omega(A_{1,0}) \times (1 - \omega(A_{2,2}))$ |
| $\top$ | $\top$ | $\top$ | $\top$ | $\omega(A_{1,2}) \times \omega(A_{2,0}) \times \omega(A_{1,0}) \times \omega(A_{2,2})$ |

In this section, we introduce in further detail some key notions used in the main body of our paper. We start with the definition of the semantic loss [56].

### A.1  Loss functions

*Semantic Loss* (SL) [56] has been adopted to train classifiers subject to logical theories [30, 43, 24]. SL is the cross entropy of the *weighted model counting* (WMC) [7] of a formula subject to a softmax output. Below, we provide the necessary notions. Let $\mathcal{Z}$ denote a set of Boolean variables. An *interpretation* $I$ of $\mathcal{Z}$ is a mapping from each $A \in \mathcal{Z}$ to true ($\top$) or false ($\bot$). Interpretation $I$ is a *model* of a Boolean formula $\varphi$, if $\varphi$ evaluates to true in $I$. We use the notation $A \in \varphi$ to denote that Boolean variable occurs in $\varphi$. By treating each Boolean variable $A$ occurring in $\varphi$ as an independent Bernoulli random variable that becomes true with probability $\omega(A)$ and false with probability $1 - \omega(A)$, the different interpretations of $\mathcal{Z}$ induce a probability distribution, where the probability $P_\varphi(I, \omega)$ of each interpretation $I$ subject to $\omega$ and $\varphi$ is given by:

$$P_\varphi(I, \omega) := \begin{cases} \prod_{A \in \varphi | I(A) = \top} \omega(A) \cdot \prod_{A \in \varphi | I(A) = \bot} (1 - \omega(A)) & I \text{ is a model of } \varphi, \\ 0 & \text{otherwise.} \end{cases} \quad (11)$$

The WMC $\mathrm{WMC}(\varphi, \omega)$ of $\varphi$ under $\omega$ then denotes *the probability of $\varphi$ being satisfied*, i.e., it evaluates to true, under $\omega$[5], and is defined as:

$$\mathrm{WMC}(\varphi, \omega) := \sum_{\text{Interpretation } I \text{ of the variables in } \varphi.} P_\varphi(I, \omega) \tag{12}$$

Assuming that each variable in $\varphi$ is associated with a class from $\mathcal{Y}$, and treating $f(x)$ as a mapping from the variables in $\varphi$ to the $f$'s scores on $x$, the semantic loss of $\varphi$ under $f(x)$ is defined as:

$$\mathrm{SL}(\varphi, f(x)) := -\log(\mathrm{WMC}(\varphi, f(x))) \tag{13}$$

We demonstrate the above concepts via Example 4.

**Example 9** (Example 4, cont'd). *Recall that the Boolean formula induced by the partial label $s = 2$ is $\varphi = (A_{1,2} \wedge A_{2,0}) \vee (A_{1,0} \wedge A_{2,2})$, where $A_{i,j}$ is a Boolean variable that is true iff the $i$-th input digit has label $j$. Given a probabilistic prediction $f(\boldsymbol{x})$, variable $A_{i,y}$ is assigned probability $f^y(x_i)$, inducing a probability mapping $\omega := \bigcup_{i \in \{1,2\}, y \in \{0,\dots,9\}} \{A_{i,y} \mapsto f^y(x_i)\}$. Table 3 shows all interpretations of $\{A_{1,2}, A_{2,0}, A_{1,0}, A_{2,2}\}$ and their corresponding probabilities subject to $\omega$ and $\varphi$. The WMC of $\varphi$ under $\omega$ is the sum of the probabilities of all the $2^4$ interpretations.*

To establish SL, each $(x_i, y))$, where $\boldsymbol{x} = (x_1, \dots, x_M)$, $i \in [M]$ and $y \in \mathcal{Y}^M$, is associated with a distinct Boolean variable $A_{i,y}$.

The nature of multi-instance PLL implies Boolean formulas in *disjunctive normal form* (DNF), where a formula is in DNF form, if it is a disjunction over conjunctions of Boolean variables or their negations, see Example 4. In fact, each Boolean formula computed out of the labels associated with a partial label is a positive DNF formula, i.e., a DNF formula where no variable occurs negatively. Using the union bound of standard probability theory and the model-based semantics of formula probabilities [20], the following result directly follows:

**Lemma 2.** *Let $\Phi_1, \dots, \Phi_K$ be $K$ positive conjunctive formulas over the set of Boolean variables $\mathcal{Z}$ and $\omega : \mathcal{Z} \mapsto (0, 1)$. Assuming that each variable $Z \in \mathcal{Z}$ is treated as an independent Bernoulli random variable that becomes true with probability $\omega(Z)$ and false with probability $1 - \omega(Z)$, the following holds:*

$$\mathrm{WMC}(\bigvee_{i=1}^{K} \Phi_i, \omega) \leq \sum_{i=1}^{K} \mathrm{WMC}(\Phi_i, \omega) \tag{14}$$

*where the equality holds when the $\Phi_i$'s are logically inconsistent, i.e., there is no interpretation of $\mathcal{Z}$ in which two or more conjunctions are simultaneously true.*

Let $\Sigma$ denote exclusiveness constraints among the variables occurring in the $\Phi_i$'s. For instance, returning back to Example 9, to align with the semantics of the classifier, only of the Boolean variables $A_{1,2}$ and $A_{1,0}$ can be true in each interpretation[6]. Then, $\mathrm{WMC}(\bigvee_{i=1}^{K} \Phi_i \wedge \Sigma, \omega) \leq \mathrm{WMC}(\bigvee_{i=1}^{K} \Phi_i, \omega)$. This is because $\Sigma$ reduces the number of models of $\varphi$.

In addition, when the $\Phi_i$'s share no Boolean variables, the following holds [20]:

$$\mathrm{WMC}(\bigvee_{i=1}^{K} \Phi_i, \omega) = 1 - \prod_{i=1}^{K} (1 - \mathrm{WMC}(\Phi_i, \omega)) \tag{15}$$

Due to (15), the top-$k$ partial loss becomes

$$\ell_\sigma^k(f(\boldsymbol{x}), s) = -\log(1 - \prod_{i=1}^{k} (1 - P_{f(\boldsymbol{x})}(\boldsymbol{y}^{(i)}))) \tag{16}$$

when the $\boldsymbol{y}^{(i)}$'s in Definition 3 share no Boolean variables.

---

[5]Notice that WMC also applies to the case where $\omega$ maps each Boolean variable to a non-negative real number. However, when restricting to $(0, 1)$, the WMC of a Boolean formula equals to its probability [20].

[6]The above can be captured via the integrity constraint $\neg(A_{1,2} \wedge A_{1,0})$.

## A.2 VC and Natarajan dimensions

We use the following definitions of VC and Natarajan dimensions, which can be found in [40].

**Definition 6** (Shattering: multiclass classifiers). *Let $\mathcal{H}$ be a space of multiclass classifiers of the form $\mathcal{X} \to \mathcal{Y}$. A set $C \subset \mathcal{X}$ is* shattered *by $\mathcal{H}$, if there exist two functions $h_0, h_1 : C \to \mathcal{Y}$, such that*

- *For each $x \in C$, we have that $h_0(x) \neq h_1(x)$.*

- *For each $B \subset C$, there exists a function $h \in \mathcal{H}$, such that*

$$\forall x \in B, h(x) = h_0(x) \quad and \quad \forall x \in C \backslash B, h(x) = h_1(x)$$

**Definition 7** (Natarajan Dimension). *The* Natarajan dimension *of $\mathcal{H}$, denoted by $\mathrm{Nat}(\mathcal{H})$, is the maximal size of a shattered set $C \subset \mathcal{X}$. If $\mathcal{H}$ is a space of binary classifiers, then its Natarajan dimension is also called its* VC dimension *and is denoted by $\mathrm{VC}(\mathcal{H})$.*

## A.3 Rademacher complexity

We use the following definition of the (empirical) Rademacher complexity of a multiclass scoring space $\mathcal{F}$, which is adapted from [10].

**Definition 8** (Rademacher complexity). *Let $\mathcal{T} = \{x_1, \ldots, x_m\}$ be a set of instances that are i.i.d. drawn from $\mathcal{D}_X$. The* empirical Rademacher complexity *of $\mathcal{F}$ with respect to $\mathcal{T}$ is defined by*

$$\widehat{\mathfrak{R}}_m(\mathcal{F}; \mathcal{T}) := \frac{1}{m} \mathbb{E}_{\boldsymbol{\epsilon}} \left[ \sup_{f \in \mathcal{F}} \sum_{i=1}^{m} \sum_{y \in \mathcal{Y}} \epsilon_{i,y} f^y(x_i) \right] \tag{17}$$

*where the $\epsilon_{i,y}$'s are i.i.d. Rademacher random variables, each of which is uniformly distributed over $\{-1, +1\}$, and $\boldsymbol{\epsilon}$ is a vector over all the $\epsilon_{i,y}$'s. Furthermore, the* Rademacher complexity *of scoring class $\mathcal{F}$ is the expectation of the empirical version:*

$$\mathfrak{R}_m(\mathcal{F}) := \mathbb{E}_{\mathcal{T} \sim \mathcal{D}_X^m}[\widehat{\mathfrak{R}}_m(\mathcal{F}; \mathcal{T})] \tag{18}$$

# B Proofs for Section 3

Before proceeding with the proofs, we will introduce some useful notation.

**Definition 9** (Probability of misclassification). *Let $f$ be a scoring function in $\mathcal{F}$ and $l_i, l_j$ be two labels in $\mathcal{Y}$. Then, the probability that $f$ misclassifies label $l_i$ as $l_j$ subject to samples drawn from $\mathcal{D}$ is defined as*

$$E_{l_i, l_j}(f) := \mathbb{P}_{(x,y) \sim \mathcal{D}}([f](x) = l_j \cap y = l_i) \tag{19}$$

For convenience, here we consider the general setting where the transition function $\sigma$ is unknown to the learner but instead, a transition class $\mathcal{G}$ is provided. The results for the special case where $\sigma$ is known to the learner can be derived by setting $\mathcal{G} = \{\sigma\}$. Now, fix a scoring function class $\mathcal{F}$ and a transition class $\mathcal{G}$. We define $\mathcal{H}_{\mathcal{F},\mathcal{G}}$ as the class of binary classifiers that, given samples of the form $(\boldsymbol{x}, s)$, for $\boldsymbol{x} \in \mathcal{X}^M$ and $s \in \mathcal{S}$, output zero, if the predictions for $\boldsymbol{x}$ abide by the partial label $s$ and one, otherwise. Formally:

$$\mathcal{H}_{\mathcal{F},\mathcal{G}} := \{(\boldsymbol{x}, s) \mapsto \mathbb{1}\{\sigma'([f](\boldsymbol{x})) \neq s\} | f \in \mathcal{F}, \sigma' \in \mathcal{G}\} \tag{20}$$

For the case where the transiton is known to the learner, we also denote $\mathcal{H}_{\mathcal{F},\mathcal{G}} = \mathcal{H}_{\mathcal{F}}$ for simplicity. From (20), it follows that the zero-one binary classification loss of a classifier $f_{\sigma'} \in \mathcal{H}_{\mathcal{F},\mathcal{G}}$, equals the zero-one partial loss subject to $\sigma'$ defined as $\ell_{\sigma'}^{01}(f(\boldsymbol{x}), s) := \mathbb{1}\{\sigma'(f(\boldsymbol{x})) \neq s\}$. We now present the following lemma that bounds the VC dimension of $\mathcal{H}_{\mathcal{F},\mathcal{G}}$, which is reminiscent of the arguments of [28, 50].

**Lemma 3.** *Fix a scoring function class $\mathcal{F}$ and a transition class $\mathcal{G}$. Let $\mathrm{Nat}([\mathcal{F}]) = d_{[\mathcal{F}]} < \infty$ be the Natarajan dimension of $[\mathcal{F}]$ and $d_{\mathcal{G}}$ be the VC dimension of the class of classifiers $\{(\boldsymbol{y}, s) \mapsto \mathbb{1}\{\sigma'(\boldsymbol{y}) \neq s\} | \sigma' \in \mathcal{G}\}$, where $\boldsymbol{y} \in \mathcal{Y}^M$ and $s \in \mathcal{S}$. The VC dimension of class $\mathcal{H}_{\mathcal{F},\mathcal{G}}$ is bounded as:*

$$d \leq 2\left((d_{[\mathcal{F}]} + d_{\mathcal{G}}) \log(6M(d_{[\mathcal{F}]} + d_{\mathcal{G}})) + 2d_{[\mathcal{F}]} \log c\right) \tag{21}$$

*Proof.* Suppose that the VC dimension of $\mathcal{H}_{\mathcal{F},\mathcal{G}}$ is $d$. Let $N$ be the maximum number of distinct ways to assign labels in $\mathcal{Y}^M$ to $d$ points in $\mathcal{X}^M \times \mathcal{S}$ using $[\mathcal{F}]$. Each possible label assignment leads to a set of $d$ elements in $\mathcal{X}^M \times \mathcal{Y}^M \times \mathcal{S}$, and for any $d$ elements, by Sauer-Shelah lemma (see, for example, [40]'s Lemma 6.10), there are at most

$$\left(\frac{\mathrm{e}Md}{d_{\mathcal{G}}}\right)^{d_{\mathcal{G}}}$$

ways for $\mathcal{G}$ to decide if $\mathbb{1}\{\sigma'(\boldsymbol{y}) \neq s\}$, where $\mathrm{e}$ is the base of the natural logarithm. Based on the above, we have:

$$2^d \leq N \left(\frac{\mathrm{e}Md}{d_{\mathcal{G}}}\right)^{d_{\mathcal{G}}}$$

Therefore, $N \geq 2^d \left(d_{\mathcal{G}}/\mathrm{e}Md\right)^{d_{\mathcal{G}}}$.

By Natarajan's lemma of multiclass classification, we have:

$$N \leq (Md)^{d_{[\mathcal{F}]}} c^{2d_{[\mathcal{F}]}} \tag{22}$$

Combining (22) with the above equations, it follows that

$$(Md)^{d_{[\mathcal{F}]}} c^{2d_{[\mathcal{F}]}} \geq N \geq 2^d \left(\frac{d_{\mathcal{G}}}{\mathrm{e}Md}\right)^{d_{\mathcal{G}}}$$

Taking the logarithm on both sides, we have:

$$d_{[\mathcal{F}]} \log(Md) + 2d_{[\mathcal{F}]} \log c \geq d \log 2 + d_{\mathcal{G}}(\log d_{\mathcal{G}} - \log(Md) - 1)$$

Rearranging the inequality yields:

$$
\begin{aligned}
d \log 2 + d_{\mathcal{G}}(\log d_{\mathcal{G}} - 1) &\leq (d_{[\mathcal{F}]} + d_{\mathcal{G}})(\log d + \log M) + 2d_{[\mathcal{F}]} \log c \\
&\leq (d_{[\mathcal{F}]} + d_{\mathcal{G}}) \left(\frac{d}{6(d_{[\mathcal{F}]} + d_{\mathcal{G}})} + \log(6(d_{[\mathcal{F}]} + d_{\mathcal{G}})) - 1\right) \\
&\quad + (d_{[\mathcal{F}]} + d_{\mathcal{G}}) \log M + 2d_{[\mathcal{F}]} \log c \\
&= d/6 + (d_{[\mathcal{F}]} + d_{\mathcal{G}}) \left(\log(6M(d_{[\mathcal{F}]} + d_{\mathcal{G}})) - 1\right) + 2d_{[\mathcal{F}]} \log c \\
&\leq d/6 + (d_{[\mathcal{F}]} + d_{\mathcal{G}}) \log(6M(d_{[\mathcal{F}]} + d_{\mathcal{G}})) - d_{\mathcal{G}} + 2d_{[\mathcal{F}]} \log c
\end{aligned}
$$

where the second step follows from the first-order Taylor series expansion of the logarithm function at the point $6(d_{[\mathcal{F}]} + d_{\mathcal{G}})$. Therefore,

$$
\begin{aligned}
d &\leq \frac{(d_{[\mathcal{F}]} + d_{\mathcal{G}}) \log(6M(d_{[\mathcal{F}]} + d_{\mathcal{G}})) + 2d_{[\mathcal{F}]} \log c - d_{\mathcal{G}}(\log(d_{\mathcal{G}}))}{\log 2 - 1/6} \\
&\leq 2 \left((d_{[\mathcal{F}]} + d_{\mathcal{G}}) \log(6M(d_{[\mathcal{F}]} + d_{\mathcal{G}})) + 2d_{[\mathcal{F}]} \log c\right)
\end{aligned}
$$

where the last step follows from the fact that $\log 2 - 1/6 > 1/2$. The above concludes the proof of Lemma 3. $\square$

Notice that if $\sigma$ is known to the learner (i.e., $\mathcal{G}$ is a singleton) and hence, $d_{\mathcal{G}} = 0$[7], then (21) becomes

$$d \leq 2 \left(d_{[\mathcal{F}]} \log(6Md_{[\mathcal{F}]}) + 2d_{[\mathcal{F}]} \log c\right) \tag{23}$$

## B.1 Proofs for Section 3.1

**Lemma 1.** *If $\sigma$ is $M$-unambigous, then we have:*

$$\mathcal{R}^{01}(f) \leq \mathcal{O}(\mathcal{R}_{\mathsf{P}}^{01}(f;\sigma)^{1/M}) \quad as \quad \mathcal{R}_{\mathsf{P}}^{01}(f;\sigma) \to 0 \tag{1}$$

*Moreover, if $\sigma$ is not $M$-unambiguous, then learning from partial labels is arbitrarily difficult, in the sense that a classifier $f$ with partial risk $\mathcal{R}_{\mathsf{P}}^{01}(f;\sigma) = 0$ can have a risk of $\mathcal{R}^{01}(f) = 1$.*

---

[7] Singleton hypothesis classes have VC-dimension equal to zero.

*Proof.* Given (19), the zero-one risk of $f$ can be rewritten as

$$\mathcal{R}^{01}(f) = \sum_{l_i \neq l_j} E_{l_i, l_j}(f) \tag{24}$$

Since $\sigma$ is $M$-unambigous, we know that if all the $M$ input instances have the same label and are wrongly classified as another label, then the predicted partial label will be wrong. Therefore, the zero-one partial risk is lower bounded by the probability of such events, namely the sum of the same type of classification mistake being repeated by $M$ times. We have that:

$$
\begin{aligned}
\mathcal{R}_\mathsf{P}^{01}(f; \sigma) &\geq \sum_{l_i \neq l_j} E_{l_i, l_j}(f)^M \\
&\geq \frac{\left( \sum_{l_i \neq l_j} E_{l_i, l_j}(f) \right)^M}{(c(c-1))^{M-1}} \quad \text{(Power Mean inequality)} \\
&= \frac{\mathcal{R}^{01}(f)^M}{(c(c-1))^{M-1}}
\end{aligned}
\tag{25}
$$

where $|\mathcal{Y}| = c$. Rearranging the inequality yields:

$$
\begin{aligned}
\mathcal{R}^{01}(f) &\leq (\mathcal{R}_\mathsf{P}^{01}(f; \sigma)(c(c-1))^{M-1})^{1/M} \\
&\leq (c^{2M-2} \mathcal{R}_\mathsf{P}^{01}(f; \sigma))^{1/M} \\
&= \mathcal{O}((\mathcal{R}_\mathsf{P}^{01}(f; \sigma))^{1/M})
\end{aligned}
\tag{26}
$$

The above concludes the first part of the proof of Lemma 1.

We now focus on the second part of the proof. If $\sigma$ is not $M$-unambiguous, then from Definition 1, we know that there exists a pair of labels $y, y' \in \mathcal{Y}$ with $y' \neq y$, such that $\sigma(y, \ldots, y) \neq \sigma(y', \ldots, y')$ holds. Consider now a sample $(x_0, y)$ drawn from $\mathcal{D}$. If $\mathcal{D}$ concentrates all its mass on $x_0 \in \mathcal{X}$, i.e., $\mathbb{P}_\mathcal{D}(x_0) = 1$, and classifier $f$ misclassifies $x_0$ as $y' \neq y$, then $\mathcal{R}_\mathsf{P}^{01}(f; \sigma) = 0$ but $\mathcal{R}^{01}(f) = 1$. The above concludes the proof of Lemma 1. $\qquad \square$

**Theorem 6** (ERM learnability under $M$-unambiguity)**.** *Suppose $\mathcal{F}$ is realizable under $\ell_\mathsf{P}^{01}$ and $[\mathcal{F}]$ has a finite Natarajan dimension $d_{[\mathcal{F}]}$. Then for any $\epsilon, \delta \in (0, 1)$, there exists a universal[8] constant $C_0 > 0$, such that with probability at least $1 - \delta$, the empirical partial risk minimizer with $\widehat{\mathcal{R}}_\mathsf{P}^{01}(f; \sigma; \mathcal{T}_\mathsf{P}) = 0$ has a classification risk $\mathcal{R}^{01}(f) < \epsilon$, if*

$$m_\mathsf{P} \geq C_0 \frac{c^{2M-2}}{\epsilon^M} \left( d_{[\mathcal{F}]} \log(6cMd_{[\mathcal{F}]}) \log\left( \frac{c^{2M-2}}{\epsilon^M} \right) + \log\left( \frac{1}{\delta} \right) \right) \tag{27}$$

*Proof.* Let $f$ be the empirical partial risk minimizer, such that $\widehat{\mathcal{R}}_\mathsf{P}^{01}(f; \sigma; \mathcal{T}_\mathsf{P}) = 0$ holds. For any $\epsilon \in (0, 1)$, we know by the proof of Lemma 1 that $\mathcal{R}^{01}(f) \leq (c^{2M-2} \mathcal{R}_\mathsf{P}^{01}(f; \sigma))^{1/M}$ holds. Hence, by bounding $\mathcal{R}_\mathsf{P}^{01}(f; \sigma) \leq \epsilon^M / c^{2M-2}$, the following would follow: $\mathcal{R}^{01}(f) \leq \epsilon$. By the standard bound for sample complexity based on VC-dimension (see, for example, Theorem 6.7 in [40]), we know that for any confidence parameter $\delta \in (0, 1)$, $\mathcal{R}^{01}(f) \leq (c^{2M-2} \mathcal{R}_\mathsf{P}^{01}(f; \sigma))^{1/M}$ holds with probability at least $1 - \delta$, if

$$m_\mathsf{P} \geq m(\mathcal{H}_\mathcal{F}, \delta, \epsilon^M / c^{2M-2}) \tag{28}$$

where

$$m(\mathcal{H}_\mathcal{F}, \delta, t) := \frac{C}{t} \left( \mathcal{H}_\mathcal{F} \log\left( \frac{1}{t} \right) + \log\left( \frac{1}{\delta} \right) \right) \tag{29}$$

where $C$ is a universal constant. Combining (23) with (28) and (29) yields the desired result, concluding the proof of Theorem 6. $\qquad \square$

Before proving Proposition 1, we will introduce a more general definition of unambiguity:

**Definition 10** ($I$-unambiguity)**.** *Mapping $\sigma$ is $I$-unambiguous if there exists a set of distinct $I$ indices $\mathcal{I} = \{i_1, \ldots, i_I\} \subseteq [M]$, such that for any $\boldsymbol{y} \in \mathcal{Y}^M$ having the same label $l$ in all positions in $\mathcal{I}$, the vector $\boldsymbol{y}'$ that results after flipping $l$ to $l' \neq l$ in those positions satisfies $\sigma(\boldsymbol{y}') \neq \sigma(\boldsymbol{y})$.*

---

[8] A constant is universal if it that does not depend on the parameters of the learning problem (e.g., $M$ or $c$).

We now prove the following lemma based on $I$-unambiguity:

**Lemma 4.** *If $\sigma$ is both $I$- and $M$-unambigous, then we have:*

$$\mathcal{R}^{01}(f) \leq \mathcal{O}((\mathcal{R}_{\mathsf{P}}^{01}(f;\sigma))^{1/I}) \tag{30}$$

*Proof.* Let $\mathcal{I}$ be the disambiguation set of $\sigma$. Then, for each pair of labels $l_i, l_j \in \mathcal{Y}$ with $l_i \neq l_j$, the following holds with probability $E_{l_i,l_j}(f)^I$: the gold labels in $\mathcal{I}$ are all $l_i$, but they are all misclassified by $f$ as $j$. Since $\sigma$ is $I$-unambiguous, this type of error implies a partial label error. Then, we have the following:

$$\mathcal{R}_{\mathsf{P}}^{01}(f;\sigma) \geq \sum_{l_i \neq l_j} E_{l_i,l_j}(f)^I \underbrace{(1 - \mathcal{R}^{01}(f))^{M-I}}_{\text{Probability of correctly classifying the remaining } M-I \text{ labels.}}$$

$$\geq \frac{\left(\sum_{l_i \neq l_j} E_{l_i,l_j}(f)\right)^I (1 - \mathcal{R}^{01}(f))^{M-I}}{(c(c-1))^{I-1}} \quad \text{(Power-Mean Inequality)} \tag{31}$$

$$= \frac{\mathcal{R}^{01}(f)^I (1 - \mathcal{R}^{01}(f))^{M-I}}{(c(c-1))^{I-1}}$$

Since $\sigma$ is also $M$-unambigous by assumption, we have from (26) that $\mathcal{R}^{01}(f) \leq c^2(\mathcal{R}_{\mathsf{P}}^{01}(f;\sigma))^{1/M}$. Hence, $1 - \mathcal{R}^{01}(f) \geq 1 - c^2(\mathcal{R}_{\mathsf{P}}^{01}(f;\sigma))^{1/M}$. Combining the above with (31), yields:

$$\mathcal{R}_{\mathsf{P}}^{01}(f;\sigma) \geq \frac{\mathcal{R}^{01}(f)^I (1 - (c^{2M-2}\mathcal{R}_{\mathsf{P}}^{01}(f;\sigma))^{1/M})^{M-I}}{(c(c-1))^{I-1}}$$

Therefore,

$$\mathcal{R}^{01}(f) \leq \left(\frac{\mathcal{R}^{01}(f)c^{2I-2}}{(1 - (c^{2M-2}\mathcal{R}_{\mathsf{P}}^{01}(f;\sigma))^{1/M})^{M-I}}\right)^{1/I}$$

Based on the above, we conclude that

$$\mathcal{R}^{01}(f) \leq \Phi_I(\mathcal{R}_{\mathsf{P}}^{01}(f;\sigma)) \tag{32}$$

where

$$\Phi_I : t \mapsto \min\left\{\left(\frac{tc^{2I-2}}{(1 - (c^{2M-2}t)^{\frac{1}{M}})^{M-I}}\right)^{1/I}, t^{\frac{1}{M}}c^2\right\} \tag{33}$$

as $\mathcal{R}_{\mathsf{P}}(f;\ell_{\mathsf{P}}^{01}) \to 0$. We can see that $\Phi_I(t) = \mathcal{O}(t^{1/I})$ as $t \to 0$. The above concludes the proof of Lemma 4. $\qquad\square$

We are now ready to prove Proposition 1.

**Proposition 1** (ERM learnability under 1- and $M$-unambiguity). *If $\sigma$ is both 1- and $M$-unambigous, then we have:*

$$\mathcal{R}^{01}(f) \leq \mathcal{O}(\mathcal{R}_{\mathsf{P}}^{01}(f;\sigma)) \quad \text{as} \quad \mathcal{R}_{\mathsf{P}}^{01}(f;\sigma) \to 0 \tag{3}$$

*Furthermore, if $\mathcal{F}$ is realizable under $\ell_{\mathsf{P}}^{01}$ and $[\mathcal{F}]$ has a finite Natarajan dimension $d_{[\mathcal{F}]}$, then for any $\delta \in (0,1)$ and $\epsilon \in (0,1)$ that is sufficiently close to 0, there exists a universal constant $C_1$, such that with probability at least $1 - \delta$, the empirical partial risk minimizer with $\widehat{\mathcal{R}}_{\mathsf{P}}^{01}(f;\sigma) = 0$ has a classification risk $\mathcal{R}^{01}(f) < \epsilon$, if*

$$m_{\mathsf{P}} \geq C_1 \frac{1}{\epsilon}\left(d_{[\mathcal{F}]}\log(6cMd_{[\mathcal{F}]})\log\left(\frac{2}{\epsilon}\right) + \log\left(\frac{1}{\delta}\right)\right) \tag{4}$$

*Proof.* The first claim can be derived from Lemma 4, by setting $I = 1$. For the second claim, using inequality (33) and letting $I = 1$, we know that

$$\mathcal{R}^{01}(f) \leq \frac{t}{(1 - (c^{2M-2}t)^{1/M})^{M-1}}$$

Suppose $\epsilon$ is small enough so that

$$\epsilon \leq \frac{1}{(2M)^M c^{2M-2}} \tag{34}$$

Now, if we could bound $\mathcal{R}_\mathsf{P}^{01}(f;\sigma) \le \epsilon/2$, we will have that

$$\mathcal{R}^{01}(f) \le \frac{\epsilon/2}{(1 - (c^{2M-2}(\epsilon/2))^{\frac{1}{M}})^{M-1}}$$

$$\le \frac{\epsilon/2}{(1 - (c^{2M-2}\epsilon)^{\frac{1}{M}})^{M}}$$

$$\le \frac{\epsilon/2}{1 - M(c^{2M-2}\epsilon)^{\frac{1}{M}}} \qquad \text{(Bernoulli's inequality)}$$

$$\le \epsilon \qquad\qquad\qquad\qquad \text{(Equation (34))}$$

Now, our goal is to bound the sample complexity so that $\mathcal{R}_\mathsf{P}^{01}(f;\sigma) \le \epsilon/2$. Similarly to the proof of Theorem 6, by the standard bound for sample complexity based on VC-dimension, for any $\delta \in (0,1)$, $\mathcal{R}_\mathsf{P}^{01}(f;\sigma) \le \epsilon/2$ holds with probability at least $1 - \delta$, if

$$m_\mathsf{P} \ge m(\mathcal{H}_{\mathcal{F},\mathcal{G}}, \delta, \epsilon/2) \tag{35}$$

where

$$m(\mathcal{H}_{\mathcal{F},\mathcal{G}}, \delta, t) := \frac{C}{t}\left(\mathrm{VC}(\mathcal{H}_{\mathcal{F},\mathcal{G}})\log\left(\frac{1}{t}\right) + \log\left(\frac{1}{\delta}\right)\right) \tag{36}$$

where $C$ is a universal constant. Combining (23) with (35) and (36) yields the desired result. The above concludes the proof of Proposition 1. $\qquad\square$

## B.2 Proofs for Section 3.2

### B.2.1 Relating the top-$k$ semantic loss to the infimum/minimum loss

In this section, we show that the minimal loss can be viewed as a special case of the top-$k$ partial loss from Definition 3.

We start by recapitulating the definition of the *minimal loss* [29]. Given a scoring function $f : \mathcal{X} \to \mathbb{R}^M$, a loss function $\ell$ and a partial labelset $s \in 2^\mathcal{Y}$, the *minimal loss* [29] for standard PLL [28] is defined as:

$$\min_{y \in s} \ell(f(x), y) \tag{37}$$

Below, we show that the minimal loss coincides with the top-1 partial loss.

For multi-instance PLL, we can extend (37) in a straightforward fashion as follows:

$$\min_{\sigma(\boldsymbol{y})=s} \ell(f(\boldsymbol{x}), \boldsymbol{y}) \tag{38}$$

If $\ell$ is the cross-entropy loss, then the minimal loss finds the most likely label vector in the preimage of $s$. Now, let us focus on Definition 3. If we set $k = 1$ in $\ell_\sigma^k$, then the only model for the formula $\varphi = \boldsymbol{y}^{(1)} := A_{1,y_1^{(1)}} \wedge \cdots \wedge A_{M,y_M^{(1)}}$ assigns to each Boolean variable occurring in $\boldsymbol{y}^{(1)}$ value $\top$. The WMC of $\varphi$ is given by

$$\prod_{i=1}^{M} f^{y_i^{(1)}}(x_i) = \max_{\sigma(\boldsymbol{y})=s} P_{f(\boldsymbol{x})}(\boldsymbol{y}^{(1)}) \tag{39}$$

which is also the maximum likelihood that can be obtained in the preimage of $s$. This implies that the top-1 semantic loss finds the minimum negative log-likelihood. Therefore, the top-$k$ partial loss equals to the minimal loss for $k = 1$.

Furthermore, for a partial label $s \in 2^\mathcal{Y}$ and a label prediction $z$, the *infimum loss* [5] is defined as

$$L(z, s) := \inf_{y \in s} \ell(z, y)$$

Treating the probabilistic prediction $f(\boldsymbol{x})$ as a generalized label, the above definition becomes equivalent to that of the minimal loss (notice that $\mathcal{Y}$ is finite, so infimum = minimum).

The above discussion shows that the minimal loss is a special case of the top-$k$ partial loss.

### B.2.2 Proofs

To prove the main result of this section, we will need the following definitions and lemmas.

**Definition 11** (Top-$k$ partial $\ell^1$ loss). *For an integer $k \geq 1$, the* top-$k$ partial $\ell^1$ loss *under scoring function $f$, input $\boldsymbol{x} \in \mathcal{X}^M$ and partial label $s \in \mathcal{S}$ is given by*

$$\tilde{\ell}_\mathsf{P}^k(f(\boldsymbol{x}), s) := 1 - \mathrm{WMC}\left(\bigvee_{i=1}^k \boldsymbol{y}^{(i)}, f(\boldsymbol{x})\right) \tag{40}$$

*where $\boldsymbol{y}^{(1)}, \ldots, \boldsymbol{y}^{(k)}$ are as in Definition 3.*

We now prove the following lemma:

**Lemma 5.** *For any classifier $f \in \mathcal{F}$, $\boldsymbol{x} \in \mathcal{X}^M$, $s \in \mathcal{S}$ and integer $k \geq 1$, we have:*

$$\ell_\mathsf{P}^{01}(f(\boldsymbol{x}), s) \leq (k+1)\tilde{\ell}_\mathsf{P}^k(f(\boldsymbol{x}), s) \leq (k+1)\ell_\mathsf{P}^k(f(\boldsymbol{x}), s) \tag{41}$$

*Proof.* Let us denote by $\boldsymbol{y}_f = ([f](x_1), \ldots, [f](x_M))$ the most likely label assignment to $\boldsymbol{x}$ by $f$. If $\ell_\mathsf{P}^{01}(\boldsymbol{y}_f, s) = 1$ holds, then $\sigma(\boldsymbol{y}_f) \neq s$. Therefore $\boldsymbol{y}_f$ is different from the top-$k$ $\boldsymbol{y}^{(i)}$ vectors. Hence, the probabilities of the $\boldsymbol{y}^{(i)}$'s, for $i \in [k]$, and $\boldsymbol{y}_f$ sum to at most 1, i.e.,

$$
\begin{aligned}
1 &\geq P_{f(\boldsymbol{x})}(\boldsymbol{y}_f) + \sum_{i=1}^k P_{f(\boldsymbol{x})}(\boldsymbol{y}^{(i)}) \\
&\geq \left(\frac{1}{k} + 1\right)\sum_{i=1}^k P_{f(\boldsymbol{x})}(\boldsymbol{y}^{(i)})
\end{aligned}
\tag{42}
$$

The second inequality holds because the assignment $\boldsymbol{y}_f$ has a larger score than any other label assignments in the preimage of $s$. Furthermore, by Lemma 2, we know that

$$\mathrm{WMC}\left(\bigvee_{i=1}^k \boldsymbol{y}^{(i)}, f(\boldsymbol{x})\right) \leq \sum_{i=1}^k P_{f(\boldsymbol{x})}(\boldsymbol{y}^{(i)}) \leq \frac{k}{k+1} \tag{43}$$

Since the zero-one loss $\ell_\sigma^{01}(\boldsymbol{y}_f, s)$ is by definition either 0 or 1, we have:

$$\tilde{\ell}_\mathsf{P}^k(f(\boldsymbol{x}), s) \geq \frac{1}{k+1} \geq \frac{1}{k+1}\ell_\mathsf{P}^{01}(f(\boldsymbol{x}), s)$$

Finally, from the inequality $-\log(1-t) \geq t \; \forall t > 0$, we know that $\tilde{\ell}_\mathsf{P}^k(f(\boldsymbol{x}), s)$ lower bounds the cross-entropy top-$k$ loss:

$$\tilde{\ell}_\mathsf{P}^k(f(\boldsymbol{x}), s) \leq \ell_\mathsf{P}^k(f(\boldsymbol{x}), s) \tag{44}$$

The above concludes the proof of Lemma 5. $\qquad\square$

The following two lemmas concern the Lipschitzness of the top-$k$ loss.

**Lemma 6** (Contraction lemma (Lemma 5 from [10])). *Let $N$ and $m$ be two positive integers. Let also $\mathcal{H}$ be a set of functions that map $\mathcal{X}$ to $\mathbb{R}^N$. Suppose that for each $i \in [m]$, function $\Psi_i : \mathbb{R}^N \to \mathbb{R}$ is $\mu_i$-Lipschitz with the 2-norm, i.e.,*

$$|\Psi_i(v') - \Psi_i(v)| \leq \mu_i\|v' - v\|_2 \quad \forall v, v' \in \mathbb{R}^N \tag{45}$$

*Then, for any set of $m$ points $x_1, \ldots, x_m \in \mathcal{X}$, the following holds:*

$$\frac{1}{m}\mathbb{E}_\sigma\left[\sup_{h \in \mathcal{H}} \sum_{i=1}^m \sigma_i \Phi_i(h(x_i))\right] \leq \frac{\sqrt{2}}{m}\mathbb{E}_\epsilon\left[\sup_{h \in \mathcal{H}} \sum_{i=1}^m \sum_{j=1}^N \epsilon_{ij}\mu_i h_j(x_i)\right] \tag{46}$$

*where the $\sigma_i$'s and the $\epsilon_{ij}$'s are independent Rademacher variables uniformly distributed over $\{-1, +1\}$.*

**Lemma 7** (Lipschitzness). *For a given scoring function $f : \mathcal{X} \times \mathcal{Y} \to \mathbb{R}$ and a vector of instances $\boldsymbol{x} \in \mathcal{X}^M$, let $f(\boldsymbol{x}) = [f^y(x_i)]_{i \in [M], y \in \mathcal{Y}} \in \mathbb{R}^{M \times |\mathcal{Y}|}$ denote the vector of scores for each label. Then, for any two scoring functions $f, f' \in \mathcal{F}$ and any sample $(\boldsymbol{x}, s) \in \mathcal{D}_\mathsf{P}$, we have:*

$$|\tilde{\ell}_\sigma^k(f(\boldsymbol{x}), s) - \tilde{\ell}_\sigma^k(f'(\boldsymbol{x}), s)| \leq \sqrt{Mk}\|f(\boldsymbol{x}) - f'(\boldsymbol{x})\|_2 \tag{47}$$

*Proof.* Given $(\boldsymbol{x}, s)$, the derivative of the WMC with respect to the score $f^y(x_i)$ for a label $y \in \mathcal{Y}$ is 0 if $A_{i,y}$ does not appear in formula $\varphi$. Otherwise,

$$
\begin{aligned}
\frac{\mathrm{WMC}(\bigvee_{i=1}^k \boldsymbol{y}^{(i)}, f(\boldsymbol{x}))}{\mathrm{d}(f^y(x_i))} &= \sum_{\text{Model } I \text{ of } \varphi} \prod_{A \in \varphi | I(A) = \top} \omega(A) \cdot \prod_{A \in \varphi | I(A) = \bot} (1 - \omega(A)) \\
&= \sum_{\text{Model } I \text{ of } \varphi} \mathcal{I}_I(A_{i,y}) \prod_{A \in \varphi_{i,y} | I(A) = \top} \omega(A) \cdot \prod_{A \in \varphi_{i,y} | I(A) = \bot} (1 - \omega(A)) \\
&\leq \sum_{\text{Model } I \text{ of } \varphi} \prod_{A \in \varphi_{i,y} | I(A) = \top} \omega(A) \cdot \prod_{A \in \varphi_{i,y} | I(A) = \bot} (1 - \omega(A)) \\
&= \frac{\mathrm{WMC}(\bigvee_{i=1}^k \boldsymbol{y}^{(i)}, f(\boldsymbol{x}))}{\mathrm{d}(f^y(x_i))} \\
&\leq 1
\end{aligned}
$$

where

$$
\mathcal{I}_I(A_{i,y}) = \begin{cases} 1 & I(A_{i,y}) = \top \\ -1 & \text{otherwise} \end{cases} \tag{48}
$$

and $\varphi_{i,y}$ is the logic formula that deletes all occurrences of $A_{i,y}$ in $\varphi$. Therefore, the 2-norm of the gradient of the mapping $f(\boldsymbol{x}) \mapsto \tilde{\ell}_{\mathsf{P}}^k(f(\boldsymbol{x}), s)$ is upper bounded by

$$
\sqrt{(\text{number of Boolean variables in } \varphi)} \leq \sqrt{Mk} \tag{49}
$$

The boundedness of the gradient implies that the function $f(\boldsymbol{x}) \mapsto \mathrm{WMC}(\bigvee_{i=1}^k \boldsymbol{y}^{(i)}, f(\boldsymbol{x}))$ is Lipschitz with a Lipschitz constant $\sqrt{Mk}$, concluding the proof of Lemma 7. $\square$

We are now ready to present the proof of the main theorem of this section.

**Theorem 2** (Error bound under unambiguity). *Let an integer $k \geq 1$ and $\delta \in (0, 1)$. If $\sigma$ is both 1- and $M$-unambiguous, then with probability at least $1 - \delta$, we have:*

$$
\mathcal{R}^{01}(f) \leq \Phi\left((k+1)\left(\widehat{\mathcal{R}}_{\mathsf{P}}^k(f; \sigma; \mathcal{T}_{\mathsf{P}}) + 2\sqrt{k}M^{3/2}\mathfrak{R}_{Mm_{\mathsf{P}}}(\mathcal{F}) + \sqrt{\frac{\log(1/\delta)}{2m_{\mathsf{P}}}}\right)\right) \tag{7}
$$

*where $\widehat{\mathcal{R}}_{\mathsf{P}}^k(f; \sigma; \mathcal{T}_{\mathsf{P}}) = \sum_{(\boldsymbol{x}, s) \in \mathcal{T}_{\mathsf{P}}} \ell_\sigma^k(f(\boldsymbol{x}), s)/m_{\mathsf{P}}$ is the empirical counterpart of* (6) *and $\Phi$ is an increasing function that satisfies $\lim_{t \to 0} \Phi(t)/t = 1$.*

*Proof.* *Bound with the partial risk.* From the proof of Lemma 4 and $I = 1$, we know that

$$
\mathcal{R}^{01}(f) \leq \Phi(\mathcal{R}_{\mathsf{P}}^{01}(f; \sigma))
$$

where

$$
\Phi : t \mapsto \min\left\{\frac{t}{(1 - (c^{2M-2}t)^{1/M})^{M-1}}, t^{\frac{1}{M}}c^2\right\} \tag{50}
$$

We can see that $\Phi$ is monotone increasing and $\lim_{t \to 0} \Phi(t)/t = 1$.

*Bound the zero-one partial risk with the top-k loss.* From Lemma 5, we know that $\ell_{\mathsf{P}}^{01}(f(\boldsymbol{x}), s) \leq (k+1)\tilde{\ell}_{\mathsf{P}}^k(f(\boldsymbol{x}), s)$. Taking expectation yields:

$$
\mathcal{R}_{\mathsf{P}}^{01}(f; \sigma) \leq (k+1)\mathcal{R}_{\mathsf{P}}^k(f; \sigma)
$$

*Bound with the empirical risk.* Given a partially labelled dataset $\mathcal{T}_{\mathsf{P}} = \{(\boldsymbol{x}_i, s_i)\}_{i=1}^{m_{\mathsf{P}}}$, by the standard Rademacher complexity bounds, we know that for each $\delta \in (0, 1)$ and each $f \in \mathcal{F}$, the following inequality holds with probability at least $1 - \delta$:

$$
\mathcal{R}_{\mathsf{P}}^k(f; \sigma) \leq \widehat{\mathcal{R}}_{\mathsf{P}}(f; \tilde{\ell}_{\mathsf{P}}^k; \mathcal{T}_{\mathsf{P}}) + 2\mathfrak{R}_{m_{\mathsf{P}}}(\mathcal{A}_{\mathcal{F}, k}) + \sqrt{\frac{\log(1/\delta)}{2m_{\mathsf{P}}}} \tag{51}
$$

where $\mathcal{A}_{\mathcal{F},k}$ is a class of functions that take as inputs elements of the form $(\boldsymbol{x}, s)$ with $\boldsymbol{x} \in \mathcal{X}^M$ and $s \in \mathcal{S}$, and is defined as follows:

$$\mathcal{A}_{\mathcal{F},k} := \left\{ (\boldsymbol{x}, s) \mapsto \tilde{\ell}_{\mathsf{P}}^k(f(\boldsymbol{x}), s) : f \in \mathcal{F} \right\} \tag{52}$$

By Lemma 6 with $\Psi_i : f(\boldsymbol{x}_i) \to \tilde{\ell}_{\mathsf{P}}^k(f(\boldsymbol{x}_i), s_i)$, and Lipschitzness (Lemma 7) we have:

$$
\begin{aligned}
\mathfrak{R}_m(\mathcal{A}_{\mathcal{F},k}) &= \frac{1}{m_{\mathsf{P}}} \mathbb{E}_{\mathcal{T}_{\mathsf{P}}} \mathbb{E}_\sigma \left[ \sup_{f \in \mathcal{F}} \sum_{i=1}^{m_{\mathsf{P}}} \sigma_i \tilde{\ell}_{\mathsf{P}}^k(f(\boldsymbol{x}_i), s_i) \right] \\
&= \sqrt{kM} \frac{1}{m_{\mathsf{P}}} \mathbb{E}_{\mathcal{T}} \mathbb{E}_\epsilon \left[ \sup_{f \in \mathcal{F}} \sum_{i=1}^{m_{\mathsf{P}}} \sum_{j=1}^{M} \sum_{y \in \mathcal{Y}} \epsilon_{ijy} f^y(x_{ij}) \right] \\
&= \sqrt{kM} M \underbrace{\frac{1}{M m_{\mathsf{P}}} \mathbb{E}_{\mathcal{T}} \mathbb{E}_\epsilon \left[ \sup_{f \in \mathcal{F}} \sum_{i=1}^{m_{\mathsf{P}}} \sum_{j=1}^{M} \sum_{y \in \mathcal{Y}} \epsilon_{ijy} f^y(x_{ij}) \right]}_{\text{Rademacher complexity with } M \times m_P \text{ instances.}} \\
&= \sqrt{k} M^{3/2} \mathfrak{R}_{M m_{\mathsf{P}}}(\mathcal{F})
\end{aligned}
\tag{53}
$$

_Bound with cross-entropy._ Finally, from the inequality $-\log(1-t) \geq t \; \forall t > 0$, we know that

$$\widehat{\mathcal{R}}_{\mathsf{P}}(f; \tilde{\ell}_{\mathsf{P}}^k; \mathcal{T}_{\mathsf{P}}) \leq \widehat{\mathcal{R}}_{\mathsf{P}}(f; \ell_{\mathsf{P}}^k; \mathcal{T}_{\mathsf{P}}) \tag{54}$$

Putting the above inequalities together yields the proof of Theorem 2. $\qquad\square$

## C   Proofs for Section 4

### C.1   On the bounded risk assumption

In Section 4, we introduced the bounded risk assumption. Recall that the assumption requires a constant $R < 1$, such that for each $i \in [n]$ and each $f \in \mathcal{F}_i$, $\mathcal{R}^{01}(f) \leq R$ holds. Below, we discuss how this assumption is achieved with a small amount of directly labeled data $\mathcal{T}_{\mathsf{L}} = \{(x_i, y_i)\}_{i=1}^{m_{\mathsf{L}}}$ i.i.d. drawn from $\mathcal{D}$.

Given $\mathcal{T}_{\mathsf{L}}$ and a trade-off parameter $\lambda > 1$, consider the following combined learning objective:

$$\widehat{\mathcal{L}}(f; \mathcal{T}_{\mathsf{P}}, \mathcal{T}_{\mathsf{L}}) := \frac{1}{m_{\mathsf{P}}} \sum_{(\boldsymbol{x}, s) \in \mathcal{T}_{\mathsf{P}}} \ell_\sigma^{01}(f(\boldsymbol{x}), s) + \frac{\lambda}{m_{\mathsf{L}}} \sum_{(x,y) \in \mathcal{T}_{\mathsf{L}}} \ell^{01}([f](x), y)$$

Suppose that there is a classifier $f_0 \in \mathcal{F}$ that can achieve a small empirical classification risk $r_0$ given by $\frac{1}{m_{\mathsf{L}}} \sum_{(x,y) \in \mathcal{T}_{\mathsf{L}}} \ell^{01}([f_0](x), y)$[9]. Then, any classifier $f_1$ with an empirical classification risk greater than $r_0 + 1/\lambda$ will be rejected by this learning objective. This is because

$$\widehat{\mathcal{L}}(f_1; \mathcal{T}_{\mathsf{P}}, \mathcal{T}_{\mathsf{L}}) \geq \lambda \left( r_0 + \frac{1}{\lambda} \right) = \lambda r_0 + 1 \geq \widehat{\mathcal{L}}(f_0; \mathcal{T}_{\mathsf{P}}, \mathcal{T}_{\mathsf{L}})$$

Therefore, all the possibly learned classifiers must have an empirical risk less than $r_0 + 1/\lambda$. Furthermore, under standard learning theory assumptions (e.g., finite VC dimension or vanishing Rademacher complexity), for any $\delta_0 \in (0, 1)$ we can typically bound the expected classification risk with probability at least $1 - \delta_0$ by

$$r_0 + 1/\lambda + B_1 \sqrt{\frac{B_2 + \log(1/\delta_0)}{m_{\mathsf{L}}}} \tag{55}$$

where $B_1, B_2$ is a constant depending on the learning problem. As long as we can have $m_{\mathsf{L}}$ that is large enough to make (55) less than 1 (i.e., _non-vacuous_), we can guarantee that our assumption holds with probability at least $1 - \delta_0$. The above justified the bounded risk assumption.

---

[9]Under the realizable assumption, we have that $r_0 = 0$.

## C.2 Proofs

Below, we introduce the analog of Lemma 3 for the multi-classifier case. Let

$$\mathcal{H}_{\mathcal{F}_1,\ldots,\mathcal{F}_n} := \{(\boldsymbol{x},s) \mapsto \mathbb{1}\{\sigma([\boldsymbol{f}](\boldsymbol{x})) \notin s\} | \boldsymbol{f} \in \mathcal{F}_1 \times \cdots \times \mathcal{F}_n\}$$

Given a training sample $(\boldsymbol{x}, s)$, a binary classifier from $\mathcal{H}_{\mathcal{F}_1,\ldots,\mathcal{F}_n}$ predicts whether the labels predictions for $\boldsymbol{x}$ are consistent with the partial label $s$. Suppose $(\boldsymbol{x}, s)$ are drawn from $\mathcal{D}_\mathsf{P}$. Then, $0$ is always the gold label of this binary classification problem. The zero-one binary classification error of $\{(\boldsymbol{x},s) \mapsto \mathbb{1}\{\sigma([\boldsymbol{f}](\boldsymbol{x})) \notin s\}$ equals the zero-one partial loss defined as $\ell_\sigma^{01}(\boldsymbol{f}(\boldsymbol{x}),s)$.

**Lemma 8.** *Let* $\mathrm{Nat}([\mathcal{F}_i]) = d_{[\mathcal{F}_i]} < \infty$ *be the Natarajan dimension of each* $[\mathcal{F}_i]$. *Then, we can bound the VC dimension of the function class* $\mathcal{H}_{\mathcal{F}_1,\ldots,\mathcal{F}_n}$ *as*

$$d \leq 4\sum_{i=1}^{n} \left( d_{[\mathcal{F}_i]} \log(M_i \cdot n \cdot d_{[\mathcal{F}_i]}) + 2d_{[\mathcal{F}_i]} \log c_i \right) \tag{56}$$

*Proof.* By Natarajan's lemma of multiclass classification, we have:

$$2^d \leq \prod_{i=1}^{n} (M_i d)^{d_{[\mathcal{F}_i]}} c_i^{2d_{[\mathcal{F}_i]}} \tag{57}$$

where $c_i = |\mathcal{Y}_i|$. Taking the logarithm on both sides, we have

$$d \log 2 \leq \sum_{i=1}^{n} d_{[\mathcal{F}_i]} \log(M_i d) + 2d_{[\mathcal{F}_i]} \log c_i$$

Rearranging the inequality yields:

$$d \log 2 \leq \sum_{i=1}^{n} \left( d_{[\mathcal{F}_i]} \left( \log(nM_i d_{[\mathcal{F}_i]}) + \frac{d}{ned_{[\mathcal{F}_i]}} \right) + 2d_{[\mathcal{F}_i]} \log c_i \right)$$

$$= \sum_{i=1}^{n} \left( d_{[\mathcal{F}_i]} \log(nM_i d_{[\mathcal{F}_i]}) + 2d_{[\mathcal{F}_i]} \log c_i \right) + \frac{d}{\mathrm{e}}$$

where the second step follows from the first-order Taylor series expansion of the logarithm function $\log(M_i t)$ at the point $t = ned_{[\mathcal{F}_i]}$. Using the fact that $(\log 2 - 1/\mathrm{e})^{-1} < 4$, yields (56), concluding the proof of Lemma 8. $\square$

**Theorem 3** (ERM learnability under multi-unambiguity). *Assume that there is a constant* $R < 1$, *such that for each* $i \in [n]$, *each* $f \in \mathcal{F}_i$ *is zero-one risk $R$-bounded. Assume also that there exist positive integers* $M^*$ *and* $c^*$, *such that* $M_i \leq M^*$ *and* $c_i \leq c_0$ *hold for any* $i \in [n]$. *Then, if* $\sigma$ *is multi-unambiguous, we have:*

$$\mathcal{R}^{01}(\boldsymbol{f}) \leq \mathcal{O}((\mathcal{R}_\mathsf{P}^{01}(\boldsymbol{f};\sigma))^{1/M^*}) \quad as \quad \mathcal{R}_\mathsf{P}^{01}(\boldsymbol{f};\sigma) \to 0 \tag{8}$$

*Furthermore, for any* $\epsilon, \delta \in (0,1)$, *there is a universal constant* $C_3$, *such that with probability at least* $1 - \delta$, *the empirical partial risk minimizer with* $\widehat{\mathcal{R}}_\mathsf{P}^{01}(f;\sigma) = 0$ *has a classification risk* $\mathcal{R}^{01}(f) < \epsilon$ *if*

$$m_\mathsf{P} \geq C_3 \frac{nc_0^{2M^*-2}}{\epsilon^{M^*}(1-R)^M} \left( \sum_{i=1}^{n} d_{[\mathcal{F}_i]} \log(nc_i M_i d_{[\mathcal{F}_i]}) \log\left( \frac{nc_0^{2M^*-2}}{\epsilon^{M^*}(1-R)^M} \right) + \log\left(\frac{1}{\delta}\right) \right) \tag{9}$$

*Proof.* Recall that $c_i = |\mathcal{Y}_i|$. By the multi-unambiguity assumption, we can lower bound the partial label risk as

$$\mathcal{R}_\mathsf{P}^{01}(\boldsymbol{f};\sigma) \geq \sum_{i=1}^{n} \underbrace{\frac{(\mathcal{R}^{01}(f_i))^{M_i}}{(c_i(c_i-1))^{M_i-1}} \prod_{j\neq i}(1-\mathcal{R}^{01}(f_j))^{M_j}}_{\text{Probability that each } x \in \boldsymbol{x}_i \text{ is misclassified but each other prediction is correct.}}$$

$$\geq \sum_{i=1}^{n} \frac{(\mathcal{R}^{01}(f_i))^{M_i}}{(c_i(c_i-1))^{M^*-1}} \prod_{j\neq i}(1-R)^{M_j}$$

$$= \sum_{i=1}^{n} \frac{(\mathcal{R}^{01}(f_i))^{M_i}}{(c_i(c_i-1))^{M^*-1}}(1-R)^{M-M_i} \tag{58}$$

$$\geq \sum_{i=1}^{n} \frac{(\mathcal{R}^{01}(f_i))^{M^*}}{(c_0(c_0-1))^{M^*-1}}(1-R)^{M-M_*}$$

$$\geq \frac{(1-R)^{M-M_*}}{(nc_0(c_0-1))^{M^*-1}} \left(\sum_{i=1}^{n}\mathcal{R}^{01}(f_i)\right)^{M^*}$$

Therefore, we have:

$$\mathcal{R}^{01}(\boldsymbol{f}) \leq \left(\frac{(nc_0(c_0-1))^{M^*-1}}{(1-R)^{M-M_*}}\mathcal{R}_\mathsf{P}^{01}(\boldsymbol{f};\sigma)\right)^{1/M^*}$$

$$\leq \left(\frac{(nc^2)^{M^*-1}}{(1-R)^{M-M_*}}\mathcal{R}_\mathsf{P}^{01}(\boldsymbol{f};\sigma)\right)^{1/M^*} \tag{59}$$

$$= \mathcal{O}((\mathcal{R}_\mathsf{P}^{01}(\boldsymbol{f};\sigma))^{1/M^*})$$

The above concludes the first part of Theorem 3.

To show the second part, from (59), we know that if the following holds:

$$\mathcal{R}_\mathsf{P}^{01}(\boldsymbol{f};\sigma) \leq \frac{\epsilon^{M^*}(1-R)^{M-M_*}}{(nc_0^2)^{M^*-1}} \tag{60}$$

then, $\mathcal{R}^{01}(\boldsymbol{f}) \leq \epsilon$ holds. By the standard bound for sample complexity based on VC-dimension, we know that for any confidence parameter $\delta \in (0,1)$, inequality (60) holds with probability no less than $1-\delta$, if

$$m_\mathsf{P} \geq m\left(\mathcal{H}_{\mathcal{F}_1,\dots,\mathcal{F}_n}, \delta, \frac{\epsilon^{M^*}(1-R)^{M-M_*}}{(nc_0^2)^{M^*-1}}\right) \tag{61}$$

where

$$m(\mathcal{H}_{\mathcal{F}_1,\dots,\mathcal{F}_n}, \delta, t) := \frac{C}{t}\left(\mathrm{VC}(\mathcal{H}_{\mathcal{F}_1,\dots,\mathcal{F}_n})\log\left(\frac{1}{t}\right) + \log\left(\frac{1}{\delta}\right)\right) \tag{62}$$

and $C$ is a universal constant. Combining (56) with (61) and (62), concludes the second part of Theorem 3. $\square$

**Theorem 4** (Error bound under multi-unambiguity with multiple classifiers). *Suppose $\sigma$ is multi-unambiguous and each $f_i$ is zero-one risk $R$-bounded, for $R \in (0,1)$. Then, for any integer $k \geq 1$ and any $\delta \in (0,1)$, with probability at least $1-\delta$, we have:*

$$\mathcal{R}^{01}(\boldsymbol{f}) \leq \left(\frac{nc_0^{2M^*-2}(k+1)}{(1-R)^M}\left(\widehat{\mathcal{R}}_\mathsf{P}^k(\boldsymbol{f};\sigma;\mathcal{T}_\mathsf{P}) + \sqrt{kM}\sum_{i=1}^{n}M_i\mathfrak{R}_{m_\mathsf{P}M_i}(\mathcal{F}_i) + \sqrt{\frac{\log(1/\delta)}{2m_\mathsf{P}}}\right)\right)^{1/M^*}$$

*Proof.* Let

$$\boldsymbol{f}(\boldsymbol{x}) = [f_i^y(x_{ij})]_{i\in[n],j\in[M_i],y\in\mathcal{Y}_i}$$

be the vector that encodes all the probabilities that $\boldsymbol{f}$ assign to a pair $(x,y)$ where $x$ is from $\boldsymbol{x}$ and $y$ is a possible label for $x$. Similarly to the single-variable case, we define

$$\tilde{\ell}_\mathsf{P}^k(\boldsymbol{f}(\boldsymbol{x}), s) := 1 - \mathrm{WMC}\left(\bigvee_{i=1}^{k}\boldsymbol{y}^{(i)}, \boldsymbol{f}(\boldsymbol{x})\right) \tag{63}$$

*Bound with the partial risk.* From the proof of Theorem 3, we know that

$$\mathcal{R}^{01}(\boldsymbol{f}) \le \left( \frac{(nc^2)^{M^*-1}}{(1-R)^{M-M_*}} \mathcal{R}_{\mathsf{P}}^{01}(\boldsymbol{f}; \sigma) \right)^{1/M^*}$$

*Bound with the top-k loss.* Similarly to the single-classifier case, it can be shown that

$$\mathcal{R}_{\mathsf{P}}^{01}(\boldsymbol{f}; \sigma) \le (k+1)\mathcal{R}_{\mathsf{P}}(\boldsymbol{f}; \tilde{\ell}_{\mathsf{P}}^k)$$

*Bound with the empirical risk.* By standard Rademacher complexity bounds, given a partially labeled dataset $\mathcal{T}_{\mathsf{P}}$ of size $m_{\mathsf{P}}$, for each $\delta \in (0,1)$ and each $f \in \mathcal{F}$, the following holds with probability at least $1 - \delta$:

$$\mathcal{R}_{\mathsf{P}}^k(\boldsymbol{f}; \sigma) \le \widehat{\mathcal{R}}_{\mathsf{P}}(\boldsymbol{f}; \tilde{\ell}_{\mathsf{P}}^k) + 2\mathfrak{R}_{m_{\mathsf{P}}}(\mathcal{H}_{\mathcal{F}}) + \sqrt{\frac{\log(1/\delta)}{2m_{\mathsf{P}}}} \tag{64}$$

where $\mathcal{H}_{\mathcal{F},k}$ is a class of functions that take as input elements of the form $(\boldsymbol{x}, s)$, with $\boldsymbol{x} \in \mathcal{X}^M$ and $s \in \mathcal{S}$, and is defined as follows:

$$\mathcal{H}_{\mathcal{F},k} := \left\{ (\boldsymbol{x}, s) \mapsto \tilde{\ell}_{\mathsf{P}}^k(\boldsymbol{f}(\boldsymbol{x}), s) : f \in \mathcal{F} \right\} \tag{65}$$

Denote $\mathcal{F} = \prod_{i=1}^M \mathcal{F}_i$. Using exactly the same argument in Lemma 7, we can show the mapping $\boldsymbol{f}(\boldsymbol{x}) \mapsto \tilde{\ell}_{\mathsf{P}}^k(\boldsymbol{f}(\boldsymbol{x}), s)$ is Lipschitz with Lipschitz constant $\sqrt{kM}$, By the contraction lemma (Lemma 6), we have that

$$
\begin{aligned}
\mathfrak{R}_m(\mathcal{H}_{\mathcal{F},k}) &= \frac{1}{m_{\mathsf{P}}} \mathbb{E}_{\mathcal{T}_{\mathsf{P}}} \mathbb{E}_{\sigma} \left[ \sup_{\boldsymbol{f} \in \mathcal{F}} \sum_{i=1}^{m_{\mathsf{P}}} \sigma_i \tilde{\ell}_{\mathsf{P}}^k(\boldsymbol{f}(\boldsymbol{x}), s) \right] \\
&= \sqrt{kM} \frac{1}{m_{\mathsf{P}}} \mathbb{E}_{\mathcal{T}} \mathbb{E}_{\epsilon} \left[ \sup_{\boldsymbol{f} \in \mathcal{F}} \sum_{i=1}^{m_{\mathsf{P}}} \sum_{l=1}^n \sum_{j=1}^{M_l} \sum_{y \in \mathcal{Y}_i} \epsilon_{iljy} f_l^y(x_{lj}) \right] \\
&\le \sqrt{kM} \sum_{l=1}^n M_l \frac{1}{m_{\mathsf{P}} M_l} \mathbb{E}_{\mathcal{T}} \mathbb{E}_{\epsilon} \left[ \sup_{\boldsymbol{f} \in \mathcal{F}} \sum_{i=1}^{m_{\mathsf{P}}} \sum_{j=1}^{M_l} \sum_{y \in \mathcal{Y}_i} \epsilon_{iljy} f_l^y(x_{lj}) \right] \\
&= \sqrt{kM} \sum_{l=1}^n M_l \mathfrak{R}_{m_{\mathsf{P}} M_l}(\mathcal{F}_l) \\
&= \sqrt{kM} \sum_{i=1}^n M_i \mathfrak{R}_{m_{\mathsf{P}} M_i}(\mathcal{F}_i)
\end{aligned} \tag{66}
$$

The above implies

$$\mathcal{R}_{\mathsf{P}}(\boldsymbol{f}; \ell_{\mathsf{P},1}^k) \le \widehat{\mathcal{R}}_{\mathsf{P}}(\boldsymbol{f}; \tilde{\ell}_{\mathsf{P}}^k; \mathcal{T}_{\mathsf{P}}) + 2\sqrt{kM} \sum_{i=1}^n M_i \mathfrak{R}_{m_{\mathsf{P}} M_i}(\mathcal{F}_i) + \sqrt{\frac{\log(1/\delta)}{2m_{\mathsf{P}}}} \tag{67}$$

*Bound with cross-entropy.* Finally, from the inequality $-\log(1-t) \ge t \ \forall t > 0$, we know that

$$\widehat{\mathcal{R}}_{\mathsf{P}}(\boldsymbol{f}; \tilde{\ell}_{\mathsf{P}}^k; \mathcal{T}_{\mathsf{P}}) \le \widehat{\mathcal{R}}_{\mathsf{P}}(\boldsymbol{f}; \ell_{\mathsf{P}}^k; \mathcal{T}_{\mathsf{P}}) \tag{68}$$

Putting the above inequalities together, the proof of Theorem 4 follows. $\qquad \square$

## D  Proofs for Section 5

**Theorem 5.** *If $\mathcal{G}$ is unambigous and any $f \in \mathcal{F}$ is $r$-bounded, then we have:*
$$\mathcal{R}^{01}(f) \le O(\mathcal{R}_{\mathsf{P}}^{01}(f; \mathcal{G})^{1/M}) \quad \text{as} \quad \mathcal{R}_{\mathsf{P}}^{01}(f; \mathcal{G}) \to 0 \tag{10}$$
*Furthermore, suppose $[\mathcal{F}]$ has a finite Natarajan dimension $d_{[\mathcal{F}]}$ and the function class $\{(\boldsymbol{y}, s) \mapsto \mathbb{1}\{\sigma'(\boldsymbol{y}) \ne s\} | \sigma' \in \mathcal{G}\}$ has a finite VC-dimension $d_{\mathcal{G}}$. Then, for any $\epsilon, \delta \in (0,1)$, there is a universal constant $C_4$ such that with probability at least $1 - \delta$, the empirical partial risk minimizer with $\widehat{\mathcal{R}}_{\mathsf{P}}^{01}(f; \sigma) = 0$ has a classification risk $\mathcal{R}^{01}(f) < \epsilon$, if*

$$m_{\mathsf{P}} \ge C_4 \frac{c^{2M-2}}{r^M \epsilon^M} \left( \left((d_{[\mathcal{F}]} + d_{\mathcal{G}}) \log(6M(d_{[\mathcal{F}]} + d_{\mathcal{G}})) + d_{[\mathcal{F}]} \log c\right) \log\left(\frac{c^{2M-2}}{r^M \epsilon^M}\right) + \log\left(\frac{1}{\delta}\right) \right)$$

*Proof.* Fix a classifier $f \in \mathcal{F}$. Since $\mathcal{G}$ is unambiguous, it follows that for each pair of labels $(l_i, l_j)$ such that $E_{l_i, l_j}(f) > 0$ and each $\sigma' \in \mathcal{G}$, there exists a vector $\boldsymbol{y} \in \{l_i, l_j\}^M$, such that $\sigma'(\boldsymbol{y}) \neq \sigma(l_i, l_i, \ldots, l_i)$ holds. Therefore, we have:

$$
\begin{aligned}
\min_{\sigma' \in \mathcal{G}} \mathcal{R}_{\mathsf{P}}^{01}(f; \sigma') &\geq \sum_{i \neq j} \min_{0 \leq k \leq M} E_{l_i, l_i}(f)^k E_{l_i, l_j}(f)^{M-k} \\
&\geq \sum_{l_i \neq l_j} r^M E_{l_i, l_j}^M(f) \\
&\geq \frac{r^M}{c^{2M-2}} \left( \sum_{l_i \neq l_j} E_{l_i, l_j}(f) \right)^M \\
&= \frac{r^M}{c^{2M-2}} \mathcal{R}^{01}(f)^M
\end{aligned}
\tag{69}
$$

The above implies that $\mathcal{R}^{01}(f) \leq O(\mathcal{R}_{\mathsf{P}}^{01}(f; \sigma)^{1/M})$ holds.

To prove the second part of Theorem 5, from the above discussion we know that $\mathcal{R}^{01}(f) \leq (c^{2M-2} \mathcal{R}_{\mathsf{P}}^{01}(f; \sigma)/r^M)^{1/M}$. Therefore, if we could control $\mathcal{R}_{\mathsf{P}}^{01}(f; \sigma) \leq \epsilon^M r^M / c^{2M-2}$, we would have that $\mathcal{R}^{01}(f) \leq \epsilon$. By the standard bound for sample complexity based on VC-dimension, then for any $\delta \in (0, 1)$, $\mathcal{R}_{\mathsf{P}}^{01}(\hat{f}; \sigma) \leq \epsilon^M r^M / c^{2M-2}$ can happen with probability at least $1 - \delta$ if

$$
m_{\mathsf{P}} \geq m(\mathcal{H}_{\mathcal{F}, \mathcal{G}}, \delta, \epsilon^M r^M / c^{2M-2})
\tag{70}
$$

where

$$
m(\mathcal{H}_{\mathcal{F}, \mathcal{G}}, \delta, t) = \frac{C}{t} \left( \mathsf{VC}(\mathcal{H}_{\mathcal{F}, \mathcal{G}}) \log\left(\frac{1}{t}\right) + \log\left(\frac{1}{\delta}\right) \right)
\tag{71}
$$

and $C$ is a universal constant. Now, recall (21) from Lemma 3. Combining (21) with (70) and (71) yields the proof of the second part of Theorem 5. $\square$

# E  Results for standard PLL

## E.1  Counterexamples with transition matrices

**Definition 12** (Transition matrix [8, 46]). *A transition matrix $\mathbf{T}$ for a learning problem with hidden label $Y \in \mathcal{Y}$ and observation $S \in \mathcal{S}$ is a stochastic matrix of dimension $|\mathcal{Y}| \times |\mathcal{S}|$, where the element in its $i^{th}$ column and $j^{th}$ row is the conditional probability $\mathbb{P}(S = j | Y = i)$. Transition matrix $\mathbf{T}$ is invertible if it is left invertible, i.e., there is a matrix $\mathbf{P}$ of dimension $|\mathcal{Y}| \times |\mathcal{S}|$ such that $\mathbf{P}\mathbf{T} = \mathbf{I}_{|\mathcal{Y}|}$.*

As shown in [8, 46], if the transition is invertible, then it is possible to construct an unbiased estimator for the classification loss using the samples of the partial label $s$. Therefore, the invertibility of the transition is desirable since it implies one can minimize the classification risk with a partially labeled dataset alone.

**A transition matrix formulation for multi-instance PLL.** Consider the multi-instance PLL with a single classifier and a deterministic transition $\sigma$. Consider a naive formulation of the transition matrix where we view $\mathcal{Y}^M$ as the label space. Then, if there are two different label assignments $\boldsymbol{y} \neq \boldsymbol{y}'$ such that $\sigma(\boldsymbol{y}) \neq \sigma(\boldsymbol{y}')$, their corresponding column vectors in $\mathbf{T}$ will be the same, resulting in a non-invertible transition. For example, in the MNIST with SUM2 problem, the column vectors corresponding to the label pairs $(0, 1)$ and $(1, 0)$ will be:

$$
\begin{array}{c}
\\
s = 0 \\
s = 1 \\
s = 2 \\
\vdots \\
s = 18
\end{array}
\begin{array}{cc}
\boldsymbol{y} = (1, 2) & \boldsymbol{y} = (2, 1) \\
\left[\begin{array}{cc}
0 & 0 \\
1 & 1 \\
0 & 0 \\
\vdots & \vdots \\
0 & 0
\end{array}\right]
\end{array}
\tag{72}
$$

This fact implies that the transition $\sigma$ must be injective to ensure invertibility, which is too strong for the partial labels in practice.

To overcome this issue, we consider an alternative formulation, where we view $s$ as a randomized partial label for each of the $M$ instances, where the randomness comes from the distribution of the other instances. Suppose the marginal distribution of $Y$ is known (or can be estimated in some ways). Then, we construct a transition matrix $\mathbf{T}_k$ for each $k \in [M]$ by defining its $i^{\text{th}}$ column and $j^{\text{th}}$ row to be $\mathbb{P}(s = j | Y_i = i) = \mathbb{P}(\boldsymbol{y} | \sigma(\boldsymbol{y}) = j, y_k = i)$. For example, the transition matrices for the MNIST with 2sum problem are

$$
\mathbf{T}_1 = \mathbf{T}_2 = 
\begin{array}{c}
\\
s=0 \\
s=1 \\
s=2 \\
\vdots \\
s=9 \\
s=10 \\
s=11 \\
\vdots \\
s=18
\end{array}
\begin{array}{cccc}
0 & 1 & \cdots & 9 \\
\left[\begin{array}{cccc}
\mathbb{P}(Y=0) & 0 & \cdots & 0 \\
\mathbb{P}(Y=1) & \mathbb{P}(Y=0) & \cdots & 0 \\
\mathbb{P}(Y=2) & \mathbb{P}(Y=1) & \cdots & 0 \\
\vdots & \vdots & \ddots & \vdots \\
\mathbb{P}(Y=9) & \mathbb{P}(Y=8) & \cdots & \mathbb{P}(Y=0) \\
0 & \mathbb{P}(Y=9) & \cdots & \mathbb{P}(Y=1) \\
0 & 0 & \cdots & \mathbb{P}(Y=2) \\
\vdots & \vdots & \ddots & \vdots \\
0 & 0 & \cdots & \mathbb{P}(Y=9)
\end{array}\right]
\end{array}
\tag{73}
$$

We say the standard PLL problem is *invertible* if there exists an $k \in [M]$ such that $\mathbf{T}_k$ is left invertible.

**$M$-unambiguity $\not\Rightarrow$ invertibility for multi-instance PLL.** Consider the following example, where $M$-unambiguity holds but the transition matrix is not invertible for no input position. Consider $\mathcal{Y} = [3]$ and $M = 3$. Assume $\sigma(1, 1, 1) \neq \sigma(2, 2, 2) \neq \sigma(3, 3, 3)$ so that $\sigma$ is $M$-unambiguous. Now, suppose

$$
\begin{aligned}
\sigma(1, 1, 1) &= \sigma(1, 2, 3) = \sigma(1, 3, 2) \\
&= \sigma(2, 1, 2) = \sigma(2, 2, 1) = \sigma(2, 3, 3) \\
&= \sigma(3, 1, 3) = \sigma(3, 2, 2) = \sigma(3, 3, 1)
\end{aligned}
\tag{74}
$$

and

$$
\begin{aligned}
\sigma(1, 1, 2) &= \sigma(1, 2, 1) = \sigma(1, 3, 3) \\
&= \sigma(2, 1, 3) = \sigma(2, 2, 2) = \sigma(2, 3, 1) \\
&= \sigma(3, 1, 1) = \sigma(3, 2, 3) = \sigma(3, 3, 2)
\end{aligned}
\tag{75}
$$

and

$$
\begin{aligned}
\sigma(1, 1, 3) &= \sigma(1, 2, 2) = \sigma(1, 3, 1) \\
&= \sigma(2, 1, 1) = \sigma(2, 2, 3) = \sigma(2, 3, 2) \\
&= \sigma(3, 1, 2) = \sigma(3, 2, 1) = \sigma(3, 3, 3)
\end{aligned}
\tag{76}
$$

so that $\mathcal{S} = \{\sigma(1, 1, 1), \sigma(2, 2, 2), \sigma(3, 3, 3)\}$. Suppose that $\mathbb{P}(Y = 1) = \mathbb{P}(Y = 2) = \mathbb{P}(Y = 3) = 1/3$. Then, it can be verified that

$$
\mathbf{T}_1 = \mathbf{T}_2 = \mathbf{T}_3 = 
\begin{array}{c}
\\
\sigma(1,1,1) \\
\sigma(2,2,2) \\
\sigma(3,3,3)
\end{array}
\begin{array}{ccc}
1 & 2 & 3 \\
\left[\begin{array}{ccc}
1/3 & 1/3 & 1/3 \\
1/3 & 1/3 & 1/3 \\
1/3 & 1/3 & 1/3
\end{array}\right]
\end{array}
\tag{77}
$$

This means that all the transition matrices are non-invertible.

**Inveritbility $\not\Rightarrow$ $M$-unambiguity for multi-instance PLL.** Suppose $\mathcal{Y} = [2]$ and $\sigma(y_1, y_2) = \mathbb{1}\{y_1 = y_2\}$. Then $M$-unambiguity does not hold since $\sigma(1, 1) = \sigma(2, 2)$. But if $\mathbb{P}(Y = 1) = 0.1 = 1 - \mathbb{P}(Y = 2)$, then

$$
\mathbf{T}_1 = 
\begin{array}{c}
\\
0 \\
1
\end{array}
\begin{array}{cc}
1 & 2 \\
\left[\begin{array}{cc}
0.9 & 0.1 \\
0.1 & 0.9
\end{array}\right]
\end{array}
\tag{78}
$$

is an invertible transition.

## E.2 Counterexamples for standard PLL

We show that for the standard PLL, the small ambiguity degree condition [28] and the invertibility of the transition matrix, see Definition 12, do not imply each other. Below, we recapitulate the small ambiguity degree condition.

**Definition 13** (Ambiguity degree [28]). *The ambiguity degree for standard PLL is defined as*

$$\gamma := \sup_{\mathcal{D}(x,y)>0, y'\neq y} \mathbb{P}_{(x,y)\sim\mathcal{D}}(y' \in \sigma(y)) \tag{79}$$

*where $\mathcal{D}(x,y)$ is the density function at $(x,y)$. We say that a PLL instance satisfies the* small ambiguity degree condition, *if $\gamma < 1$.*

**Small ambiguity $\not\Rightarrow$ invertibility.** Consider the following transition matrix (where all blank elements are zero) with $\mathcal{Y} = [5]$ and $\mathcal{S} = 2^{\mathcal{Y}}$:

$$\mathbf{T} = \begin{array}{c} \\ \{1,2,3\} \\ \{1,4,5\} \\ \{2,5\} \\ \{3,4\} \end{array} \begin{array}{ccccc} 1 & 2 & 3 & 4 & 5 \\ \left[\begin{array}{ccccc} 1/2 & 1/2 & 1/2 & & \\ 1/2 & & & 1/2 & 1/2 \\ & 1/2 & & & 1/2 \\ & & & 1/2 & 1/2 \end{array}\right] \end{array} \tag{80}$$

It can be verified that the small ambiguity degree is $1/2$ but this matrix is of rank $4 < 5$ and the invertibility condition does not hold.

**Invertibility $\not\Rightarrow$ small ambiguity.** Consider the following transition matrix with $\mathcal{Y} = [2]$ and $\mathcal{S} = 2^{\mathcal{Y}}$:

$$\mathbf{T} = \begin{array}{c} \\ \{1\} \\ \{1,2\} \end{array} \begin{array}{cc} 1 & 2 \\ \left[\begin{array}{cc} 1 & 0 \\ 0 & 1 \end{array}\right] \end{array} \tag{81}$$

We can see that the small ambiguity degree is $1$ but the above matrix is of full rank.

## E.3 $M$-ambiguity and small ambiguity

We show that our proposed $M$-unambiguity, see Definition 1, is an extension of the small ambiguity degree, see Definition 13. To do this, we consider a generalized setting, where $s$ is a randomized function of $\boldsymbol{y}$. In particular, similarly to standard PLL (aka superset learning problem) [28, 8], we define $\sigma$ to be a random function $\mathcal{Y}^M \to 2^{\mathcal{Y}^M}$ so that $s \in 2^{\mathcal{Y}^M}$ is a random set of vectors in $\mathcal{Y}^M$. Below, let $\mathcal{D}^M$ be a distribution followed by $M$ i.i.d. random variables $(X_i, Y_i)$, where $(X_i, Y_i) \sim D$, for each $i \in [M]$. We use $\mathcal{D}^M(\boldsymbol{x}, \boldsymbol{y})$ to denote the density at $(\boldsymbol{x}, \boldsymbol{y})$, for $\boldsymbol{x} \in \mathcal{X}^M$ and $\boldsymbol{y} \in \mathcal{Y}^M$.

**Definition 14** ($M$-Ambiguity degree). *The ambiguity degree for standard PLL subject to $\mathcal{D}^M$ is given by*

$$\gamma := \sup_{\mathcal{D}^M(\boldsymbol{x},\boldsymbol{y})>0, \boldsymbol{y},\boldsymbol{y}' \text{ are diagonal with } \boldsymbol{y}\neq\boldsymbol{y}'} \mathbb{P}_{(\boldsymbol{x},\boldsymbol{y})\sim\mathcal{D}^M}(\boldsymbol{y}' \in s) \tag{82}$$

*We say that a PLL instance satisfies the* small $M$-ambiguity degree condition, *if $\gamma < 1$.*

Firstly, it can be seen that the $M$-ambiguity degree, see Definition 14, reduces to the ambiguity degree from Definition 13 when $M = 1$, since all label vectors are diagonal in that case. Below, we show that the small $M$-ambiguity degree condition guarantees learnability.

**Proposition 2.** *If the small $M$-ambiguity degree condition is satisfied, then we have:*

$$\mathcal{R}^{01}(f) \leq \frac{1}{1-\gamma}(c^{2M-2}\mathcal{R}_{\mathsf{P}}^{01}(f;\sigma))^{1/M} \tag{83}$$

*Proof.* Let $\mathcal{E}$ be the event that for an input vector $\boldsymbol{x} \in \mathcal{X}^M$ with gold labels $\boldsymbol{y}$, there exists a diagonal vector $\boldsymbol{y}' \in \mathcal{Y}^M$ with $\boldsymbol{y}' \neq \boldsymbol{y}$, such that $\boldsymbol{y}' \in s$. Then, by definition, we have $\mathbb{P}(\mathcal{E}) \leq \gamma$. Also, conditioned on $\mathcal{E}$, from the proof of Lemma 1, we know that $\mathcal{R}_{\mathsf{P}}^{01}(f) \geq \frac{\mathcal{R}^{01}(f)^M}{(c(c-1))^{M-1}}$. The above implies that

$$\mathcal{R}_{\mathsf{P}}^{01}(f;\sigma) \geq (1 - \mathbb{P}(\mathcal{E}))\frac{\mathcal{R}^{01}(f)^M}{(c(c-1))^{M-1}} \tag{84}$$

Therefore, we have:

$$\mathcal{R}^{01}(f) \leq \frac{\left(\mathcal{R}_{\mathsf{P}}^{01}(f;\sigma)c^2\right)^{1/M}}{1 - \mathbb{P}(\mathcal{E})} \leq \frac{\left(\mathcal{R}_{\mathsf{P}}^{01}(f;\sigma)c^2\right)^{1/M}}{1 - \gamma}$$

concluding the proof of Proposition 2. $\qquad\qquad\qquad\qquad\qquad\qquad\qquad\qquad\qquad\square$

The special case of Proposition 2, where $M = 1$ recovers the classical result for standard PLL (see the discussion at the end of Section 3 in [28]). The partial risk $\mathcal{R}_{\mathsf{P}}^{01}(f;\sigma)$ can be further bounded by its empirical counterpart, assuming that the Natarajan dimension is finite, as we have previously shown.

# F    Related Work

**Partial Label Learning.** Partial label learning (aka *superset learning*) assumes the learner receives the training data in the form of $(x, z)$, where $z \in 2^{\mathcal{Y}}$ is a subset of the label space that contains the gold label. Technically, it is technically possible to cast our problem to standard PLL by identifying $s$ as its preimages, namely the set of all labels that is consistent with the observation $s$. Nevertheless, our work extends the standard PLL in two ways. Firstly, our framework allows multiple instances to appear in one training sample. Secondly, the learnability of PLL is typically shown under the *small ambiguity* assumption, see, for example, [11, 28, 5]. This condition requires no distracting label to be contained in $z$ with probability 1, which is violated in our case since a distracting label can *always* occur because $\sigma$ is deterministic. Therefore, our proposed conditions relax and generalize small ambiguity by allowing deterministic transitions, see Appendix E.3.

Another line of work in weak supervision exploits the *transition matrices* [46] to compute posterior class probabilities in different scenarios, e.g., PLL [8, 9] and noisy label learning [61]. The key assumption is that the transition matrix is *invertible* [8, 46]. Our learnability condition, $M$-unambiguity, and the invertibility condition do not imply each other, see Appendix E.1 and E.2 for a self-contained discussion. There, we also show that small ambiguity and the invertibility condition do not imply each other, which might be of independent interest.

**Latent structure models.** Our work also relates to *latent structural learning*, where the aim is to learn a latent representation using supervision on a deterministic transition $\sigma$ over the latent variables [45, 52]. To reduce the cost of computing $\sigma$, [42] proposes approximations for a restricted class of decomposable transitions. Both [42, 35] assume latent models in the exponential family. Our work is also related to SPIGOT [34, 32]. There, the aim is to learn both the transition and the latent model, proposing techniques for back-propagating through the argmax layer of the latent model. Complementary to the above research, we focus on rigorous theoretical analysis.

Our work is closely related to [62], which studies a similar problem of weakly supervised learning with multiple instances. However, our results are stronger and closer to the aims of weak supervision in comparison to the results in [62]. This is because our work proposes sufficient and necessary conditions to recover the hidden labels (i.e., $Z_{1:K}$ in [62]), while consistency in [62] concerns the likelihood of the observed labels rather than the hidden ones, as defined in Definition 3 and 4 of [62], where two parameters are equivalent if they have the same likelihood on the observed partial label (rather than the hidden labels). Also, we use a different surrogate loss based on semantic loss that allows a top-$k$ approximation. In general, semantic loss is designed to capture Boolean constraints over the hidden labels and is standard to use in neuro-symbolic learning. Furthermore, our work directly extends the theory of PLL, which is thoroughly discussed in Appendix. In particular, our learnability condition from Definition 1 directly extends the classical small ambiguity degree condition, as shown in Appendix E.3. In contrast, the connection to PLL is missing from [62].

**Constrained learning.** The problem of partial label learning is closely related to constrained learning, in the sense that the prediction for the label (vector) is subject to the constraint $\sigma(\boldsymbol{y}) = s$ at the training stage. Training classifiers under constraints has been well studied in NLP. The work in [36] proposes a formulation for training under linear constraints; [37] proposes a learning framework that unifies expectation maximization with integer linear programming. Posterior regularization has been proposed for training classifiers under graphical models [21]. The latter two lines of research were adopted by [27] and [23] for neurosymbolic learning.

**Neurosymbolic learning.** Our work was motivated by frameworks that employ logic for training neural models [30, 53, 13, 59, 43, 31, 24, 27]. We prove learnability under (unknown) transitions that capture different languages: systems of Boolean formulas, non-linear constraints and logical theories. Notice that the key to model logical theories, e.g., Datalog, via $\sigma$ is through *abduction* [26], see [43, 13] for a discussion. We also prove error bounds under a common neurosymbolic loss, the SL [56], and top-$k$ approximations. We are the first to provide rigorous results on neurosymbolic learning, closing a gap in the literature.

Notice that [58] also proves consistency of a top-$k$ loss. However, they use a zero-one-style loss. Finally, [3] provides generalization bounds of the *smart predict-then-optimize* (SPO) loss [15]; it assumes, though, that the gold labels of the inputs are given. Complementary to our research is the work in [19, 47, 33] that aims to integrate combinatorial solvers as differentiable layers in deep neural models. Finally, [48] proposes techniques for learning to solve combinatorial optimization problems using end-to-end neural models–loosing the ability to extract the latent models.

A subtle, yet substantial, difference between standard PLL and multi-instance PLL is that populating $\sigma$ may incur a substantial computational overhead in the latter case. For instance, let us return back to Example 1. To populate the entries of $\sigma$ for each target sum $s$, we need to compute all solutions of the equation $X + Y = s$, where $X, Y \in \{0, \ldots, 9\}$. In contrast, the set of labels associated with each input instance is part of the input [28, 11].

**Learning logical theories.** Another relevant strand of research is that of learning logical theories [12, 16, 38]. The work in [12] mixes abduction with induction to jointly learn a theory while training the neural classifiers, while [16] introduces a differentiable formulation of inductive logic programming that relies on the semantics of fuzzy logic for interpreting formulas. Finally, the authors in [38] propose learning rules using the notion of logical neural networks. The techniques in [12, 16, 38] rely on templates for learning logical theories. Instead of relying on pre-specified templates, the authors in [18] aim to learn those templates by mining patterns (aka motifs) from the data, following mathematically rigorous techniques and providing formal guarantees related to the mined motifs. Differently from our work, the above line of research provides no guarantees in terms of learnability or error bounds.

# G  Experiment Details

Firstly, recall that in all neurosymbolic frameworks considered in Section 6, we considered weights in $\{1, \ldots, 5\}$[10] and used an additional neural classifier to learn the unknown weights. As the weights are fixed, the inputs to the weight classifiers in all frameworks are constant signals, i.e., in each iteration, we provided the classifier with the same signal $W_i$ for the $i$-th weight. For each neurosymbolic framework, our implementation built upon the sources made available by the authors.

We now provide information about DeepProbLog, NeurASP and NeuroLog. Listings 1, 2 and 3 show the logic programs used to train the MNIST digit classifiers under the above three frameworks using the WEIGHTED-SUM problem, see Section 6.

DeepProbLog and NeurASP link the classifiers' predictions with the logic program via *neural predicates*. WEIGHTED-SUM requires two neural predicates: digit and weight. The first one classifies the input MNIST digits into $\{0, \ldots, 9\}$, while the second one classifies the input weights into $\{1, \ldots, 5\}$. In NeuroLog, the association between the classifiers' predictions and the logic program is maintained via *abducible* predicates. In particular, the instances of those predicates associate the probabilistic predictions with the facts from the domain of the logic program. Listing 3 uses two abducible predicates: atdigit and atweight. Atom atdigit($s, j$) denotes that the $j$-th input to the digit classifier maps to value $s$. The abducible atom atweight($s, j$) is analogously defined.

In terms of heuristics, in DeepProbLog, we used the Geometric Mean [31].

Following ENT's official implementation, we encoded WEIGHTED-SUM(2) using the Z3 SMT solver. In addition, similarly to the handwritten formula evaluation scenario in [27] (Section 5.1 from [27]), we used two projection operators. To generate the initial random labels, we projected out all the digit labels and kept only the weight labels. During random walks, we projected out the second

---

[10]In NeurASP, we used weights in $\{1, \ldots, 10\}$, since due to a bug in the most recent NeurASP implementation that was provided by the authors, the output domains of all classifiers should be of the same size.

input digit's labels, keeping the labels of the first digit and the labels of both input weights. The rest of the hyperparameters, as well as the two-stage training process, were set as in the hand-written formula evaluation scenario from [27].

**Neural architectures.** The layers of the MNIST digit classifier are as follows: *Conv2d(1, 6, 5)*, *MaxPool2d(2, 2)*, *ReLU(True)*, *Conv2d(6, 16, 5)*, *MaxPool2d(2, 2)*, *ReLU(True)*, *Linear(16 * 4 * 4, 120)*, *ReLU()*, *Linear(120, 84)*, *ReLU()*, *Linear(84, N)*, *Softmax(1)*.

The neural classifiers for all frameworks but ABL were built using PyTorch 2.0.0 and Python 3.9. For ABL, we aligned with the implementation provided by the authors and used Tensorflow 2.11.0 and Keras 2.11.0. For each framework, we used the hyperparameters proposed by the authors in analogous scenarios.

```
nn(mnist_net,[X],Y,[0,1,2,3,4,5,6,7,8,9]) :: digit(X,Y).
nn(weight_net,[X],Y,[1,2,3,4,5]) :: weight(X,Y).

weighted_sum2(I1, I2, W1, W2, S) :-
    digit(I1,D1), weight(W1,A1),
    digit(I2,D2), weight(W2,A2),
    S is A1*D1+A2*D2.

weighted_sum3(I1, I2, I3, W1, W2, W3, S) :-
    digit(I1,D1), weight(W1,A1),
    digit(I2,D2), weight(W2,A2),
    digit(I3,D3), weight(W3,A3),
    S is A1*D1+A2*D2+A3*D3.

weighted_sum4(I1, I2, I3, I4, W1, W2, W3, W4,S) :-
    digit(I1,D1), weight(W1,A1),
    digit(I2,D2), weight(W2,A2),
    digit(I3,D3), weight(W3,A3),
    digit(I4,D4), weight(W4,A4),
        S is A1*D1+A2*D2+A3*D3+A4*D4.
```

Listing 1: DeepProbLog program for the WEIGHTED-SUM problem.

```
nn(digit(1,X), [0,1,2,3,4,5,6,7,8,9]) :- img(X).
nn(weight(1,X), [1,2,3,4,5]) :- wei(X).
img(i1). img(i2). img(i3). img(i4).
wei(o1). wei(o2). wei(o3). wei(o4).

weighted_sum2(X1,X2,X3,X4,S) :-
    digit(0,X1,N1), weight(0,X3,W1),
    digit(0,X2,N2), weight(0,X4,W2),
    S=W1*N1+W2*N2.

weighted_sum3(X1,X2,X3,X4,X5,X6,S) :-
    digit(0,X1,N1), weight(0,X4,W1),
    digit(0,X2,N2), weight(0,X5,W2),
    digit(0,X3,N3), weight(0,X6,W3),
    S=W1*N1+W2*N2+W3*N3.

weighted_sum4(X1,X2,X3,X4,X5,X6,X7,X8,S) :-
    digit(0,X1,N1), weight(0,X5,W1),
    digit(0,X2,N2), weight(0,X6,W2),
    digit(0,X3,N3), weight(0,X7,W3),
    digit(0,X4,N4), weight(0,X8,W4),
    S=W1*N1+W2*N2+W3*N3+W4*N4.
```

Listing 2: NeurASP program for the WEIGHTED-SUM problem.

```prolog
abducible(atdigit(_,_)).
abducible(atweight(_,_)).
digit(N) :- N in 0..9.
weight(N) :- N in 1..5.

weighted_sum2(S) :- digit(D1), atdigit(D1,1),
                    digit(D2), atdigit(D2,2),
                    weight(S1), atweight(S1,1),
                    weight(S2), atweight(S2,2),
                    S #= (D1 * S1) + (D2 * S2).

weighted_sum3(S) :- digit(D1), atdigit(D1,1),
                    digit(D2), atdigit(D2,2),
                    digit(D3), atdigit(D3,3),
                    weight(S1), atweight(S1,1),
                    weight(S2), atweight(S2,2),
                    weight(S3), atweight(S3,3),
                    S #= (D1 * S1) + (D2 * S2) + (D3 * S3).

weighted_sum4(S) :- digit(D1), atdigit(D1,1),
                    digit(D2), atdigit(D2,2),
                    digit(D3), atdigit(D3,3),
                    digit(D4), atdigit(D4,4),
                    weight(S1), atweight(S1,1),
                    weight(S2), atweight(S2,2),
                    weight(S3), atweight(S3,3),
                    weight(S4), atweight(S4,4),
                    S #= (D1 * S1) + (D2 * S2) + (D3 * S3) + (D4 * S4).
```

Listing 3: NeuroLog program for the WEIGHTED-SUM problem.

