# OpenReview forum: "On Learning Latent Models with Multi-Instance Weak Supervision"
_NeurIPS.cc/2023/Conference — NeurIPS 2023 poster_

### Official Review · Reviewer_KpA9 · 2023-06-25

**Soundness:** 4 excellent
**Presentation:** 4 excellent
**Contribution:** 3 good
**Rating:** 7
**Confidence:** 4

**Summary:**

The authors define and study the problem of multi-instance partial label learning (PLL), where weak supervision is given in the form of a (potentially) unknown transition function $\sigma$, which maps the ground truth labels onto some label set $S$.

Under this problem setting, the paper goes on to show multiple theoretical contributions
* 1) Under a known transition function (i.e. SUM2 in the MNIST task), a sufficient and necessary condition for learnability is M-ambiguity.
To get faster rates (removing the exponential term of 1/M), they show a stronger condition of 1-ambiguity (or that the output of the transition function given one label perturbation will also be changed).
They generalize this to consider a top-k loss that is a more efficient surrogate
* 2) They study the setting of learning multiple classifiers with shared label spaces, and provide similar learnability results under analogies of M-ambiguity for multiple classifiers, with an additional assumption on the boundedness of the risk function.
* 3) Finally, they provide a similar result for learning a single classifier under an unknown transition function, which requires a new assumption about the transition space being unambiguous and the boundedness of the risk function.

The paper also provides experiments to evaluate the quality of existing weakly supervised/neuro-symbolic architectures, which support their theoretical findings and takeaways.


**Strengths:**

The paper provides comprehensive theoretical results for the PLL setting. They provide results for multiple different problem setups, considering known/unknown transition functions and learning a single/multiple classifiers.

They provide rigorous theoretical analysis, which requires (in my opinion) rather intuitive assumptions.

Relevent experiments are nicely descriptive and verify their theoretical analysis.


**Weaknesses:**

As mentioned in the paper, scalability seems to be an issue with the experimental setting. On what seems to be a not overly complex task (a weighted sum of 4 MNIST digits), performance indeed significantly drops, which calls into question the widespread potential of the PLL setting (and not this paper in particular).

While the examples are given that illustrate when the various ambiguity assumptions are violated or not violated, a more in depth discussion in the case of Definition 5 would be appreciated, as this is the most interesting setting of an unknown transition function.


**Questions:**

Are the scalability issues as observed in Table 2 primarily due to the chosen neuro-symbolic methods, the PLL setting, or a combination of both factors?

**Limitations:**

Limitations are sufficiently addressed.

---

> ### Author Rebuttal · Authors · 2023-08-09
>
> > Q: As mentioned in the paper, scalability seems to be an issue with the experimental setting. On what seems to be a not overly complex task (a weighted sum of 4 MNIST digits), performance indeed significantly drops, which calls into question the widespread potential of the PLL setting (and not this paper in particular). Are the scalability issues as observed in Table 2 primarily due to the chosen neuro-symbolic methods, the PLL setting, or a combination of both factors?
>
> The scalability issue is caused by both the PLL problem and neuro-symbolic methods. Please check the section **Comments on scalability and accuracy** in our general rebuttal at the top for a detailed explanation.
>
> About the potential of multi-instance PLL and its applications, please see the **Importance of multi-instance PLL** section in the general rebuttal. Overall, we understand the criticism towards the accuracy and the scalability of the current neuro-symbolic learning techniques. However, as we discuss in the response to all the reviewers, we see these issues as an opportunity to develop new neuro-symbolic learning techniques (especially, when the transition functions are unknown) and explore new research questions. We will add this discussion in the revision of our work.
>
> > Q: While the examples are given that illustrate when the various ambiguity assumptions are violated or not violated, a more in depth discussion in the case of Definition 5 would be appreciated, as this is the most interesting setting of an unknown transition function.
>
> Thanks for the comments. We will follow your suggestion and provide a more in-depth discussion in the revised version. Below, please find a summary of clarifications.
> - About the intuition. This unambiguity notion requires that for each candidate transition $\sigma’$ and each diagonal label vector, flipping one of the labels leads to a different partial label $s$. This ensures that even if the learner chooses a wrong transition, it is still possible to detect the classification error by observing the partial error.
> - What happens when the transition space is ambiguous? If $\mathcal{G}$ is not unambiguous, then a wrong transition and an imperfect classifier may lead to zero partial risk. In other words, a wrong transition may “hide” the classification mistakes.
> - How to examine ambiguity in practice? Although in practice the true transition is hidden, one can examine this condition by checking a sufficient condition: for each $\sigma’ \in \mathcal{G}$ and each two different labels $\\{l_i \ne l_j\\} $, the set $\\{\sigma’(y) | y \in \\{l_i,l_j\\}^M \\}$ is not a singleton. The latter condition ensures that when given a fixed diagonal label vector, and when the classifier makes mistakes, the predicted partial labels are not unique, and hence cannot all agree with the ground truth label.

---

> > ### Comment · Reviewer_KpA9 · 2023-08-11
> > **Reviewer response**
> >
> > Thanks for your detailed response! I appreciate the additional discussion on scalability and the importance of the problem setting; I'll keep my score the same and wait to hear responses from other reviewers.

---

### Official Review · Reviewer_vhBR · 2023-07-04

**Soundness:** 3 good
**Presentation:** 3 good
**Contribution:** 3 good
**Rating:** 6
**Confidence:** 3

**Summary:**

This paper studies the questions of learnability and generalization under an interesting form of supervision feedback: namely, when the true label of interest is not observed, but the learner instead has access to the output of a "transition function" $\sigma(y_1, \ldots, y_M)$ computed on the labels of an $M$-tuple of examples.
The paper gives necessary and sufficient conditions on the transition function for learnability from two "partial" (evaluated on the aggregate/partial labels) loss functions: partial 0/1 loss and a surrogate for the "semantic loss" studied in other related literature. These results are extended in multiple directions and various details such as convergence rates are also derived.


**Strengths:**

- The results are well presented and the key condition (M-unambiguity) is shown in Appendix E.3 to be a natural extention of the "small ambiguity degree" assumption from Partial Label Learning (PLL) literature.

- The authors connect the convergence rate (in terms of sample complexity) of learning to the degree of unambiguity of the transition function.

- The results provide a theoretical basis for learning with the top-$k$ approximate _semantic loss_, which was introduced by other work.

- The core results are extended in multiple directions, including to the case where there are multiple different classifiers and to the case where the transition function is unknown (under additional assumptions).

- Studies and provides a nice set of learnability results for an interesting case of weak supervision that could be practically relevant.



**Weaknesses:**

- Core technical contributions / difficulties are not explained enough (e.g., Lemma 1 seems like key result for Thm 1, and the rest follows from standard tools?) What was the key technical challenge to proving the generalization results, and why is it novel / original?

- It's not clear what practical insights are gained from the results. The experiments section seems like a proof of concept---no new method other than the "baselines" is proposed or evaluated based on the theory. The main insight seems to be the qualitative relationship between $M$ and the bound, but $M$ is not really a controllable parameter in practice.

- Related to the previous point about the experiments, this setup is missing a compelling application. What's a more realistic practical scenario where learning from this type of supervision is relevant? I.e., where observing $s$ for an $M$-tuple of examples is much more practical than observing $y$ for each example?


**Questions:**

In the Remark starting at L172 the authors suggest a possible connection between the perturbation stability of the transition function and learnability / convergence rate. Could this be connected in any way to the algorithmic stability results (e.g., Bousquet and Ellisseef)? Such a connection might lead to better convergence rates?

- Am I correct that the PAC failiure probabilities $\delta$ are w.r.t. the sampling of $\mathcal{D}_P$ (not $\mathcal{D}$)? It might be good to clarify this.

A few questions I had while reading the intro/setup, some of which were cleared up later by the main text, but suggest that clarity/flow could be improved:

L117 Learnability. What is a "partial learning algorithm"?

L138 should be $\sigma_(y,\ldots,y) \ne \sigma(y',\ldots,y')$ ?

L182 why do we need to "populate $\sigma$" to compute the zero-one loss? I don't fully follow. In the sum-M case, can't we just sum the outputs predicted by $f(x_i)$ and then compare to $s$? Or is the point that this is problematic if we wanted to use a *surrogate loss* such as cross entropy, so we'd need to form a distribution over the image of $\sigma$?


**Limitations:**

The authors describe one of the main limitations of their work (the somewhat strict assumptions required on the transition function $\sigma$ due to the worst-case distributions they consider). However, the discussion could be expanded to mention broader limitations, such as the current lack of practical insights to glean from the theoretical results.

---

> ### Author Rebuttal · Authors · 2023-08-09
>
> > Q: Core technical contributions / difficulties are not explained enough (e.g., Lemma 1 seems like key result for Thm 1, and the rest follows from standard tools?) What was the key technical challenge to proving the generalization results, and why is it novel / original?
>
> Thanks for the comment. We will clarify the above in the revision of our work. Below, we provide some clarifications.
>
> Theorem 1 builds upon several non-trivial and original intermediate results that must be shown before applying the standard VC theory:
> - For empirical error: A lower bound of the classification risk (Lemma 1).
> - For generalization error: A bound of the VC dimension for the multi-instance partial label predictor (Lemma 3).
> - Counterexamples for arguing the necessity of our proposed learning condition (last paragraph of the proof of Theorem 1).
>
> The proof of Theorem 2 is more involved: it relies on a variant of the Rademacher complexity from [3] and the following original results:
> - Lemma 1.
> - An inequality that bounds the top-k loss with the zero-one loss (Lemma 5), which requires the construction of an intermediate l1 loss (Definition 11).
> - Lipschtness of the semantic loss (Lemma 7). That result is further combined with a contraction lemma from [8] (Lemma 6) to bound the Rademacher complexity of the model.
>
> > Q: It's not clear what practical insights are gained from the results. The experiments section seems like a proof of concept---no new method other than the "baselines" is proposed or evaluated based on the theory. The main insight seems to be the qualitative relationship between M and the bound, but M is not really a controllable parameter in practice.
>
> Indeed, our work does not aim to propose new algorithms. Instead, it focuses on theoretically analyzing existing methods. However, the practical insights do not limit to the qualitative relationship with $M$. The main takeaways of our bounds for practitioners also include the following:
> - The learning difficulty (rate of convergence) is characterized by the “ambiguity degree” of the transition $\sigma$ (i.e., if the transition is $M$-/$I$-unambiguous), which can be determined without training. (Theorem 1 and Proposition 1).
> - The impact of choosing the approximation strength $k$ is not always monotone. In general, the choice leads to a tradeoff between approximation error and estimation error (Theorem 2).
>
> > Q: Related to the previous point about the experiments, this setup is missing a compelling application. What's a more realistic practical scenario where learning from this type of supervision is relevant? I.e., where observing s for an M-tuple of examples is much more practical than observing y for each example?
>
> A compelling application of our formulation is visual question answering [22], where the observation $s$ (answer) is a function of multiple hidden variables $y$ (object types and their relations). The partially labeled data $(x,s)$ is used to train a classifier for learning the mapping $x \mapsto y$. The work in [22] shows that models trained via multi-instance PLL achieve better accuracy than state-of-the-art neural end-to-end architectures.
>
> Learning latent structures from downstream observations is also of great interest in NLP [AdditionalRef1] (e.g., learning semantic parsers from sentiment classification tasks), since it has been shown to achieve better model accuracy while offering interpretability. Please also check the section **importance of multi-instance PLL** in our general rebuttal above for further comments.
>
> > Q: In the Remark starting at L172 the authors suggest a possible connection between the perturbation stability of the transition function and learnability / convergence rate. Could this be connected in any way to the algorithmic stability results (e.g., Bousquet and Ellisseef)? Such a connection might lead to better convergence rates?
>
> Thanks for pointing this out. We agree that there could be deeper connections between these two types of instabilities. Intuitively, a more stable transition will lead to a more stable partial loss and a more stable PLL algorithm. We will explore this direction in future versions of our work.
>
> > Q: Am I correct that the PAC failure probabilities $\delta$ are w.r.t. the sampling of $\mathcal{D}_P$ (not $\mathcal{D}$)? It might be good to clarify this.
>
> Yes, it is correct. We will clarify this in the revised version.
>
> > Q: L117: Learnability. What is a "partial learning algorithm"?
>
> A partial learning algorithm is one that takes the partially labeled data $\mathcal{T}_P$ as input and outputs a classifier in $\mathcal{F}$. We will improve the writing to make this definition clear.
>
> > Q: L138: should be $\sigma(y,\dots,y) \ne \sigma(y’,\dots,y’)$
>
> It’s a typo, thanks for pointing it out.
>
> > Q: L182: why do we need to "populate $\sigma$" to compute the zero-one loss? I don't fully follow. In the sum-M case, can't we just sum the outputs predicted by $f(x_i)$ and then compare to $s$? Or is the point that this is problematic if we wanted to use a surrogate loss such as cross entropy, so we'd need to form a distribution over the image of $\sigma$?
>
> Is the latter. Populating $\sigma$ is required only when computing the surrogate loss in Section 3.2. We will improve the presentation to make it clearer from the context.
>
> ---
>
> - [AdditionalRef1] Backpropagating through Structured Argmax using a SPIGOT: Peng et al., ACL 2018.

---

### Official Review · Reviewer_6VmD · 2023-07-06

**Soundness:** 2 fair
**Presentation:** 2 fair
**Contribution:** 3 good
**Rating:** 5
**Confidence:** 3

**Summary:**

This paper studies a weakly supervised learning scenario where supervision signals are given to sets of instances (instead of individual instances), while the goal is still to predict labels of unseen individuals. For example, the learner is provided with a dataset in which each training example comprises a set of instances $(x_1, x_2)$ (for which the gold labels $y_1 = 1$ and $y_2 = 2$ are unobservable) and an aggregate signal $s=3$ (which is calculated by a known function $s=\sigma(y_1,y_2)=y_1 + y_2$).

The authors present learnability results for this scenario under certain assumptions. One main result is that if the function $\sigma$ has *M-unambiguity*, a perfect classifier can be learned as the number of training data approaches infinity. The authors further extend the results to situations where the function $\sigma$ is unknown.

**Strengths:**

- This paper concerns an important problem setting called *multi-instance Partial Label Learning* (multi-instance PLL), which is an extension of the standard Partial Label Learning (PLL) problem.
- The authors introduce the notion of "M-unambiguity" and show its utility in proving the learnability of the considered problem scenario under specific assumptions.
- The authors further propose the notions of "multi-unambiguity" and "unambiguous transition space" to tackle the scenarios of learning multiple classifiers and unknown transition functions.

**Weaknesses:**

- The theoretical results presented in this submission have limited significance. Generally, learnability means that a hypothesis class is learnable under any data distribution. However, the authors state that they prove learnability under distributions that concentrate mass on a single instance or label. This assumption can be invalidated in the real world. This submission primarily considers a “toughest" distribution that concentrates mass on a single instance, implying all the instances in the training data are the same. Such a distribution is unrealistic in real-world applications. Consequently, positive results under this assumption are not so meaningful. The authors may need to explore more realistic data distributions.

- Theoretical aspects of the problem setting, which is called multi-instance Partial Label Learning in this submission, have previously been examined in the literature of weakly supervised learning, as seen in [1]. This setting is known as *learning from aggregate observations*. In [1], the consistency of learners is investigated, and the expected log-likelihood in [1] is close to the semantic loss studied in this submission. To avoid confusion and potential overlap, it is essential for the authors to discuss and elucidate the distinctions between these studies.

- The terminology "Multi-Instance Partial-Label Learning" has been used in prior research, such as in [2]. Although [2] examines a different problem setting than this paper, to avoid confusion, it would be beneficial for the authors to adopt a distinct term when referring to the problem setting considered in this submission.

- The proof of Lemma 1 appears to be convoluted. Specifically, the first inequality in Equation (26) warrants further clarification. It is not immediately evident why the assertion $R_{P}^{01}(f;\sigma) \ge \sum_{l_i\neq l_j} E_{l_i,l_j}(f)^M$ holds true. In a similar vein, could you explain the first inequality of Eq. (32) in the proof of Lemma 4? Do these Lemma rely only on the M-unambiguous condition?



[1] Zhang, Y., Charoenphakdee, N., Wu, Z., & Sugiyama, M. (2020). Learning from aggregate observations. Advances in Neural Information Processing Systems, 33, 7993-8005.

[2] Tang, W., Zhang, W., & Zhang, M. L. (2022). Multi-instance partial-label learning: Towards exploiting dual inexact supervision. arXiv preprint arXiv:2212.08997.

**Questions:**

- Is the expected log-likelihood discussed in [1] equivalent to the semantic loss (or the top-k surrogate loss) examined in this submission?

- Could you provide a clearer explanation for the proof of Lemma 1, particularly regarding the first inequality in Equation (26)?  Could you also elaborate on the first inequality in Equation (32) in the proof of Lemma 4? Is it correct to state that the proofs of these lemmas solely rely on the M-unambiguous condition?

**Limitations:**

The limitation of the submission is that it relies on assumptions and conditions, such as a concentration of mass on a single instance or label, which may not align with realistic data distributions and weaken the applicability and persuasiveness of the results.

---

> ### Author Rebuttal · Authors · 2023-08-09
>
> > Q: The theoretical results presented in this submission have limited significance. The authors state that they prove learnability under distributions that concentrate mass on a single instance or label. This assumption can be invalidated in the real world. ... Consequently, positive results under this assumption are not so meaningful. The authors may need to explore more realistic data distributions.
>
> We kindly point out that there seems to be a misunderstanding about our results. Our ambiguity conditions in Definitions 1, 4 and 5 *do* apply to any probability distribution, i.e., if these ambiguity conditions are met, then we can learn under any distribution of the training data. The only reason we mentioned concentrated distributions is to prove that the proposed conditions are necessary (instead of just being sufficient). To put it differently, we show learnability under the toughest distributions in order to ensure learnability under any distribution. Such an approach of considering though cases is commonly adopted in PLL and general ML theory, e.g., in the classical theoretical PLL theory [26]. The reviewer can also check the definition of learnability in the last paragraph before Section 3.1. We apologize for any confusion caused and will improve our presentation to emphasize the generality of our results.
>
> Practically, deriving general learning conditions from the concentrated distributions is still meaningful. It shows that if the real-world data is not balanced but close to being concentrated on very few instances/classes, then PLL can be more difficult. Furthermore, we would like to note that our conditions are satisfied by many neuro-symbolic tasks, as discussed in Examples 2 & 3.
>
> > Q: Theoretical aspects of the problem setting have previously been examined in the literature, as seen in [1]. This setting is known as learning from aggregate observations. In [1], the consistency of learners is investigated, and the log-likelihood in [1] is close to the semantic loss studied in this submission.
>
> Thanks for pointing out [1]. We agree that the learning scenario is similar, yet our contributions remain valid. Our differences with [1] are shortly summarized below:
>
> - Our results are stronger and closer to the aims of weak supervision in comparison to the results in [1]. This is because our work proposes sufficient and necessary conditions to recover the hidden labels (i.e., $Z_{1:K}$ in [1]), while consistency in [1] concerns the likelihood of the observed labels rather than the hidden ones: as defined in Definition 3 & 4 of [1], where two parameters are equivalent if they have the same likelihood on the observed partial label (rather than the hidden labels).
> - The surrogate losses are different. The log-likelihood loss in [1] is different from the semantic loss in Section 3.2 of our paper. Furthermore, we allow a top-k approximation for the surrogate loss, see Section 3.2. The difference is further explained in a separate response below.
> - Our work directly extends the theory of PLL, which is thoroughly discussed in Appendix E through F. In particular, our learnability condition from Definition 1 directly extends the classical small ambiguity degree condition, as shown in Appendix E.3. In contrast, the connection to PLL is missing from [1].
>
> We will follow the suggestion to include a comparison to [1] in our revision.
>
> > Q: The terminology "Multi-Instance Partial-Label Learning" has been used in prior research, such as in [2].
>
> Thanks for pointing it out. We will consider a better name to reduce ambiguity.
>
> > Q: Is the expected log-likelihood discussed in [1] equivalent to the semantic loss (or the top-k loss) examined in this submission?
>
> No, they are different losses. To illustrate, consider Example 9 in our submission, where the induced Boolean formula is $\varphi = \{(A_{1,2} \wedge A_{2,0}) \vee (A_{1,0} \wedge A_{2,2})\}$ (recall that $A_{i,j}$ is a Boolean variable that is true iff the $i$-th input digit has label $j$). The negative log-likelihood in [1] is the sum of the probability of the two label vectors, i.e.,
> $-\log(w(A_{1,2}) \times w(A_{2,0}) + w(A_{1,0}) \times w(A_{2,2}))$, where $w(\cdot)$ denotes the confidence of the neural classifier for the corresponding prediction. However, the semantic loss is given by $-\log(1 – (1 - w(A_{1,2}) \times w(A_{2,0})) \times (1 - w(A_{1,0}) \times w(A_{2,2})))$, i.e., the sum of the right column of Table 3.
>
> In general, semantic loss is designed to capture Boolean constraints over the hidden labels and is standard to use in neuro-symbolic learning. Notice also that our work additionally allows a top-$k$ approximation of the semantic loss, as opposed to [1] which only considers the exact likelihood.
>
> > Q: Could you provide a clearer explanation for the proof of Lemma 1, particularly regarding the first inequality in Equation (26)? Could you also elaborate on the first inequality in Equation (32) in the proof of Lemma 4? Is it correct to state that the proofs of these lemmas solely rely on the M-unambiguous condition?
>
> The first inequality in (26): By $M$-unambiguity, we know that if all the $M$ input instances have the same label and are wrongly classified as another label, then the predicted partial label will be wrong. Therefore, the zero-one partial risk is lower bounded by the probability of such events, namely the sum of the same type of classification mistake being repeated by $M$ times.
>
> The first inequality in (32): Similarly, by $I$-unambiguity, we know that if $I$ input instances in $\mathcal{I}$ have the same label and are wrongly classified as another label while all the remaining $M-I$ instances are correctly classified, then the predicted partial label will be wrong. Therefore, the zero-one partial risk is lower bounded by the RHS, which computes the probability of such events.
>
> Yes, it is correct to say that the proof of these lemmas solely relies on the $M$-unambiguity and/or $I$-unambiguity.

---

> > ### Comment · Reviewer_6VmD · 2023-08-21
> >
> > Thanks for the detailed clarification. Most of my concerns have been addressed, particularly regarding the differences between this work and prior works. I hope the authors can clarify the distinction and consider using a different term to avoid ambiguity.
> > Following are my thoughts: The M-unambiguity condition proposed in this work seems both reasonable and useful, though it results in a rather slow convergence rate; The additional 1-unambiguity condition leads to a better convergence rate but seems less practical than M-unambiguity. This dilemma in this work remained unaddressed.
> > Despite this weakness, the contributions of this work seem solid.
> > Thus, I would like to increase my score from 4 to 5.

---

> > > ### Author Response · Authors · 2023-08-21
> > >
> > > Thanks for your feedback and reconsideration. Regarding the additional comment on unambiguity conditions: Firstly, we think that 1-unambiguity is still a practical learnability condition since it can be satisfied by many transition functions in neuro-symbolic learning, such as $M$-SUM (as shown in Example 3) and SORT, which maps a list of numbers to the sorted list. Secondly, to address the dilemma, we have proposed $I$-unambiguity in Definition 10 to relax the learnability condition while achieving moderate convergence rates. Given these reasons, we still believe in the theoretical and practical importance of our proposed unambiguity conditions.

---

### Official Review · Reviewer_a7AK · 2023-07-06

**Soundness:** 3 good
**Presentation:** 2 fair
**Contribution:** 3 good
**Rating:** 6
**Confidence:** 2

**Summary:**

The paper connects latent structural learning and neuro-symbolic integration and provides the first theoretical study of multi-instance PLL with possibly an unknown transition. Under such weakly supervision scenario where the transition is deterministic, it defines the necessary and sufficient condition, the minimal distribution assumptions, for the ERM-learnability of the problem. They also derive the Rademacher-style error bounds using the top-k semantics loss. The empirical results support their theory.


**Strengths:**

1. The study provides the first theoretical analysis of multi-instance PLL with possibly an unknown transition, which is a significant contribution to the learning theory community.
2. The theory holds for both known and unknown transition cases, and the assumptions are necessary and sufficient. The experiment results also support the learnability of classifiers and the conclusion that when $M$ gets larger, the learning process gets harder.

**Weaknesses:**

1. The author better move some discussion of the necessity to let the transition function be deterministic to the introduction part to prevent confusion since it seems too strict compared with the commonly used assumption.
2. The empirical results are monotone for the weighted sum transition function, and the authors should add some other empirical results for more known transition function cases.

**Questions:**

1. The scalability in this scenario is poor as Table 2, and it seems to be worse than the error bound given by the theory. Will this happen for the known transition function case? What may be the possible reasons?


**Limitations:**

The author mentions that scalability may be an important issue. No further negative societal impact raises.

---

> ### Author Rebuttal · Authors · 2023-08-09
>
> > Q: The author better move some discussion of the necessity to let the transition function be deterministic to the introduction part to prevent confusion since it seems too strict compared with the commonly used assumption.
>
> We mainly focus on deterministic transitions, since our work was motivated by neuro-symbolic learning and NLP. Notice that while looking more restrictive, learning under deterministic transitions is actually more challenging, as prior learnability assumptions [4,9,26] do not further apply. We believe it is not difficult to extend our results to randomized transitions. In fact, in Appendix E.3, we present such an extension to randomized transitions when $M=1$. We will follow the suggestion to add a relevant discussion in the introduction.
>
> > Q: The empirical results are monotone for the weighted sum transition function, and the authors should add some other empirical results for more known transition function cases.
>
> Our aim was to offer a rigorous theoretical analysis of multi-instance PLL. Hence, we would like to kindly point the reviewer to prior art that presents results for learning under known transitions, e.g., NLog [41], NASP [53] and Scallop [22], that offer a more in-depth empirical analysis. For instance, [41] presents results on learning the pieces of a chessboard via supervision on the status of the kings; [53] presents results on learning an MNIST classifier using valid SUDOKU boards; and [22] presents results on visual question answering using multi-instance partial labels.
>
> Nevertheless, to contrast learning under known vs learning under unknown transitions, we have added new results for a variant of the WEIGHTED-SUM scenario that assumes known weights. Please see the **Updates on the empirical analysis** section at the top of our general rebuttal.
>
> > Q: The scalability in this scenario is poor as Table 2, and it seems to be worse than the error bound given by the theory. Will this happen for the known transition function case? What may be the possible reasons?
>
> For known transitions, there is still the issue of scalability as already pointed out by prior art. However, it will be milder, as it requires learning fewer parameters. Please check the section **Updates on the empirical analysis** at the top to see our new results on learning with known weights.
>
> The scalability issue is caused both by the problem itself (see our discussion at top general rebuttal about WEIGTHED-SUM) and the limitations of the current neuro-symbolic methods. For instance, Scallop [22] adopts more sophisticated grounding techniques and approximations that allow it to scale to larger scenarios than DLog. Please also check the section **Comments on scalability and accuracy** in our general rebuttal above for a detailed explanation.
>
> > Q: No further negative societal impact raises.
>
> Thanks for pointing out this. We will add a discussion on broader impacts in the revised version.

---

> > ### Comment · Reviewer_a7AK · 2023-08-14
> >
> > Thank you for your detailed reply, especially for the updated empirical results and scalability. I'm not very familiar with this topic, but I think the theory results for multi-instance PLL are solid and will be helpful for relevant research. I will keep my score and wait to discuss with other reviewers.

---

### Author Rebuttal · Authors · 2023-08-10

We would like to express our gratitude to all the reviewers for their valuable feedback. Below, we address some commonly raised issues.

**Comments on the importance of multi-instance PLL raised by reviewers vhBR & KpA9**. Multi-instance PLL captures neuro-symbolic learning and latent structural learning in NLP [AdditionalRef1]. The latter settings have, in turn, several advantages over end2end neural architectures. One obvious advantage is the ability to extract and reuse the latent model. Another advantage is improved end-task accuracy. For instance, learning via multi-instance PLL can lead to architectures with higher accuracy than that of end2end architectures in NLP [AdditionalRef1] and visual QA tasks [22]. It is worth noting that several recently proposed neuro-symbolic works are based on multi-instance PLL (e.g., [11,22,25]), so we do believe that this learning setting will find more applications in the future.

**Comments on our contributions raised by reviewer 6VmD**. We would like to point out a misunderstanding regarding the applicability of our results. Reviewer 6VmD claims that we prove learnability with distributions that concentrate mass on a single instance/label. However, this is not true, as our learnability results *do* apply to any distribution, not only just the concentrated ones. The only reason we mentioned concentrated distributions is to prove that the proposed conditions are necessary (instead of just being sufficient). We will improve our presentation in the revision to emphasize the generality of our results. We also provide a more detailed discussion in the individual response below.

**Comments on scalability and accuracy made by reviewers a7AK & KpA9**. First, while WEIGHTED-SUM might seem an easy scenario at first sight, it is quite challenging, as the hidden space grows exponentially in the domain of the weights, i.e., $M^{10} \times M^5$ (recall that the weights are in $\\{1, …, 5\\}$). Notice also that when restricting to binary weights, WEIGHTED-SUM reduces to learning Boolean formulas with conjunction and disjunction [35] as discussed in Section 6. It is worth stressing that both scalability and accuracy significantly improved when the transition is known (see the **Updates on empirical analysis** section below), i.e., basically the original learning setting supported by the neuro-symbolic techniques considered in Section 6.

Second, scalability is a known problem in neuro-symbolic learning. This is partially due to the fact that the relevant problems in logical reasoning are intractable. For instance, semantic loss [50] (adopted by DLog [29], DLog-A [30], NLog [41] and Scallop [22]) requires computing the models of Boolean formulas, which is #P-complete. Reasoning over logical theories can be also intractable [AdditionalRef2] (notice that NASP fails to support the WEIGHTED-SUM scenario for $M \ge 3$ due to its inability to ground the relevant theory), if not undecidable, while various satisfiability problems are NP-hard to solve in an exact fashion. Scalability also straightforwardly affects accuracy, e.g., NLog obtains better accuracy over DLog-A for $M=3$ (see Table 2) due to its ability to explore the whole search space. The above challenges bring up new research directions.

It is worth noting that there have been promising attempts recently to tackle those challenges. For instance, the following works can be incorporated into neuro-symbolic learning techniques to improve scalability:
- A scalable grounder for probabilistic Datalog: E. Tsamoura et al., “Probabilistic Reasoning at Scale: Trigger Graphs to the Rescue”, SIGMOD 2023.
- A scalable approximate model counting technique: Mate Soos et al., “Engineering an Efficient Approximate DNF-Counter”, IJCAI 2023.

**Updates on the empirical analysis**. To further address some of the reviewers’ concerns, we repeated the WEIGHTED-SUM scenario (Section 6) using Scallop [22]. Scallop supports the top-k semantic loss (Section 3.2). In the attached PDF, the LHS of Figure 1 shows the accuracy of the MNIST digit classifier every 1000 iterations for $k = \\{1, 2, 3, 4\\}$, when $M = 4$, $m_P = 10000$. The digit classifier has been pre-trained using 16 labeled MNIST digits, having classification accuracy equal to 58%. In contrast to DLog-A and NLog (see Table 2 in our submission), the accuracy of the digit classifier improves to 84% when $k > 1$, showing that multi-instance partial labels can improve the classification accuracy even with unknown transitions. The above result empirically shows the benefits of the top-k semantic loss approximation (Section 3.2) over approximations proposed by other neuro-symbolic techniques (see Table 2 in our submission).

To address the comment of reviewer a7AK on learning with known transitions, we repeated the experiment described in the previous paragraph except that the weights are now known and randomly chosen from $\\{1, …,100\\}$ (RHS of Figure 1 in the PDF). In contrast to the unknown case, the classification accuracy exceeds 98%. To further assess scalability, we repeated the latter experiment for $M = \\{6, 10\\}, k = \\{1, 3\\}$ and $m_P = 10000$, see Figure 2. Again, the accuracy of the digit classifier exceeds 98%, a result that is quite surprising, given that the supervision signal is rather weak when $M=10$. The above shows that PLL with known transitions is, expectedly, simpler.

Finally, we want to mention that the NASP [53] authors informed us about a bug in their implementation, due to which the weight classifier was not trained. We repeated WEIGHTED-SUM for $M = 2$, obtaining accuracy up to 98%. However, even after the bug fix, NeurASP could not scale for $M \ge 3$.

We proceed with other individual comments below.

---
- [AdditionalRef1] Peng et al., “Backpropagating through Structured Argmax using a SPIGOT”, ACL 2018.
- [AdditionalRef2] J. Baget, et al., “Walking the complexity lines for generalized guarded existential rules”, IJCAI 2011.

---

### Decision · Program_Chairs · 2023-09-21

**Decision:**

Accept (poster)

**Comment:**

This papers studies a set of problems in a variant of weak supervision. In these cases, there is no access to ground-truth labels. Instead there is a known or unknown function that takes in a subset of datapoints and outputs some value that may or may not provide signal. The authors study learnability and produce error bounds in both the known and unknown signal function settings. While predominantly a theoretical paper, the authors also provide some empirical validation for their claims.

This is a nice paper that provides the fundamentals for a type of weak supervision that connects to neurosymbolic learning. Given that there is a lot of interest in these settings, but relatively little study of the basic learning-theoretical fundamentals, I believe it is a worthwhile contribution. The reviewers agree and generally agreed.